# EPiC: Efficient Video Camera Control Learning with Precise Anchor-Video Guidance

**Zun Wang** [1]  **Jaemin Cho** [2][3]  **Jialu Li** [1]  **Han Lin** [1]  **Jaehong Yoon** [4]  **Yue Zhang** [1]  **Mohit Bansal** [1]

Project page: `https://zunwang1.github.io/Epic`

## Abstract

Recent approaches for video generation with camera control often create anchor videos (i.e., rendered videos that approximate desired camera motions) to guide diffusion models as a structured prior, by rendering from estimated point clouds following camera trajectories. However, errors in point cloud and camera trajectory estimation often lead to inaccurate anchor videos with higher training cost and low efficiency, as the model is forced to compensate for rendering misalignments. To address these limitations, we introduce EPiC, an efficient and precise camera control learning framework that constructs well-aligned training anchor videos without the need for camera pose or point cloud estimation. Concretely, we create highly precise anchor videos by masking source videos based on first-frame visibility, which ensures strong alignment, eliminates the need for camera/point cloud estimation, and thus can be readily applied to any in-the-wild video. Furthermore, we introduce Anchor-ControlNet, a lightweight module that integrates anchor video guidance in visible regions to pretrained video diffusion models, with less than 1% of additional parameters. EPiC achieves efficient training with substantially fewer parameters, training steps, and less data, and generalizes robustly to anchor videos made with point clouds at test time, enabling precise 3D-informed camera control. EPiC achieves SoTA performance on RealEstate10K and MiraData for I2V camera control task. EPiC also exhibits strong zero-shot generalization to video-to-video (V2V) scenarios.

[1]University of North Carolina, Chapel Hill [2]Johns Hopkins University [3]Allen Institute for AI [4]Nanyang Technological University, Singapore. Correspondence to: Zun Wang <zunwang@cs.unc.edu>.

*Proceedings of the 43rd International Conference on Machine Learning*, Seoul, South Korea. PMLR 306, 2026. Copyright 2026 by the author(s).

## 1. Introduction

Recent advancements in video diffusion models (VDMs) (Bar-Tal et al., 2024; Girdhar et al., 2023; Hong et al., 2022; Khachatryan et al., 2023; Wang et al., 2023; Zhang et al., 2024b; Blattmann et al., 2023; Kondratyuk et al., 2023) have significantly improved the generation of realistic videos. As video generation becomes more practical, controlling the process has become a crucial requirement. A key research focus is controlling camera trajectories (Bai et al., 2025a; Yu et al., 2025a; Ren et al., 2025; Shi et al., 2024), which is essential for applications like film recapturing and virtual cinematography. Recent approaches (Ren et al., 2025; Yu et al., 2025a; Cao et al., 2025; Zhang et al., 2024a; Yu et al., 2024b) achieve this by using 3D-informed guidance to create an *'anchor video,'* which approximates the desired camera motion to guide the diffusion model. This method faces challenges, however, as it requires high-quality 3D data from expensive motion-capture systems or relies on inaccurate 3D point cloud/camera trajectory estimators (Wang et al., 2024c; Yang et al., 2024a; Schönberger et al., 2016). These inaccuracies result in pixel-level misalignments between anchor and source videos, which in turn cause training difficulties and inefficiencies (Yu et al., 2025a; 2024b), often requiring extensive computational resources and substantial backbone modifications. Furthermore, most training data mainly comes from multi-view datasets of static scenes (Zhou et al., 2018a; Ling et al., 2024) to ensure high-quality estimations, limiting the models' ability to generalize to real-world dynamic videos (Rockwell et al., 2025).

To address these issues, we propose EPiC, for learning **E**fficient and **P**recise V**i**deo **C**amera control by crafting precisely-aligned training anchor videos with a lightweight, region-aware ControlNet model design (Sec. 4). Our key insight is that anchor videos should be well-aligned with the source videos to make learning as easy, transforming the task from one of more difficult repairing misaligned content to the simpler task of copying visible regions. Thus, unlike previous approaches that render anchor videos from inaccurate 3D point clouds, which are often misaligned with the source video and reliant on camera trajectories, we directly

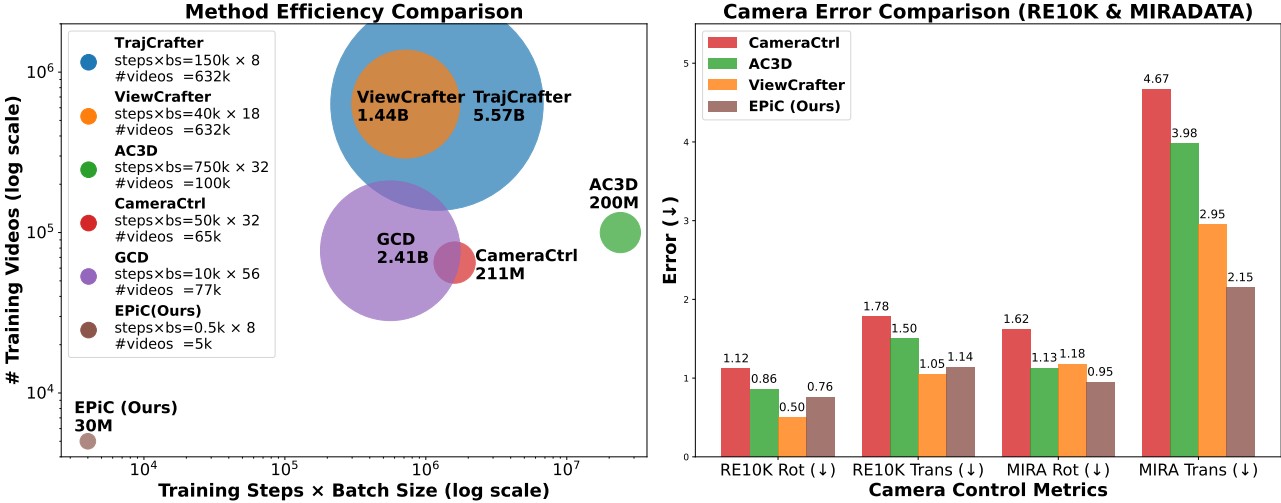

*Figure 1.* Left: Method efficiency comparison. The circle area is proportional to the number of trainable parameters (exact values are shown below method names). Our method achieves over an order of magnitude higher efficiency in terms of training data, compute cost (steps × batch size), and parameter count. Right: Camera control performance comparison. On both RealEstate10K and Mira datasets, our method achieves the best results with the lowest rotation and translation errors.

synthesize anchor videos by masking the source video based on first-frame visibility. Specifically, for each subsequent frame, we estimate its pixel trajectories with respect to the first frame from dense optical flow (Teed & Deng, 2020), preserving only those pixels that can be reliably traced back to the first frame. Pixels with no valid correspondence in the first frame are masked out. This process effectively mimics the key property of anchor videos that all new regions relative to the first frame are invisible, while ensuring precise alignment in visible regions. Additionally, our method eliminates the need for camera trajectory estimations during training, allowing anchor videos to be created from any in-the-wild source. At test time, we leverage standard point-cloud rendering to construct anchor videos for user-specified camera trajectories.

Furthermore, we introduce Anchor-ControlNet (Sec. 4.2), a method that injects anchor-video-based control signals into the generation process with the base model frozen, unlike previous anchor-video-based methods(ViewCrafter (Yu et al., 2024b), Gen3C (Ren et al., 2025) and TrajectoryCrafter (Yu et al., 2025a)) that require extensive full fine-tuning of the backbone. Anchor-ControlNet is a lightweight module with only 26M parameters (<1% of the backbone), injected into the first 25% of backbone layers and using merely 8% of the hidden dimension, directly taking the anchor video as control signals. Importantly, to improve quality in invisible regions, we introduce a novel design that makes Anchor-ControlNet visibility-aware by applying visibility masking to its outputs. Specifically, its output is added to the base model's latent representation only within the visible regions, leaving the unseen areas untouched. This design simplifies the ControlNet's task to copying visible

content, while delegating the synthesis of occluded or invisible regions entirely to the base diffusion model. This clear division of responsibility prevents errors in invisible regions from influencing the output video, reducing training difficulty and fully unleashing the base model's generative ability in unseen areas. Moreover, restricting ControlNet to visible regions naturally allows user-controlled regional motion—masks on the anchor video can indicate which regions can be moved—thus supporting both static and dynamic scene generation under the same camera trajectory at test time. Combining all these components, we show camera control can be learned with remarkable efficiency: converging with just 5K in-the-wild videos and 500 training steps (less than 5% of the data and steps of prior methods) (Fig. 1 Left), requiring only 15 GPU hours.

Extensive experiments demonstrate that, despite being over an order of magnitude more efficient, EPiC achieves superior performance in camera accuracy (*e.g.*, RotErr, TransErr; Fig. 1, Right) and motion stability (measured by the standard deviation of generated trajectories across different seeds) on I2V camera control tasks in both indoor and game environments. Moreover, EPiC exhibits strong generalization to V2V camera control in a zero-shot setting, even though it is trained solely on I2V data. Ablation study shows the effectiveness of our anchor video method and ControlNet design. Our contributions are as follows:

- A novel anchor video construction pipeline with visibility-based masking that produces well-aligned anchor–source video pairs without requiring point cloud or camera trajectory estimation during training, enabling learning from diverse in-the-wild videos.

- A lightweight Anchor-ControlNet with visibility-aware output masking, allowing efficient and precise anchor-video conditioning, as well as selective regional motion control at test time.
- Strong performance on both I2V and V2V camera control tasks with high efficiency in training, data, and model size compared to previous methods.

## 2. Related Work

**Image/Text-Based Camera Control in VDMs.** Controlling camera trajectories in text-to-video (T2V) generation and I2V generation has recently received increasing attention. A common approach is to inject explicit camera parameters (e.g., plücker Embedding) into VDMs (Wang et al., 2024e; Hou et al., 2024; Bahmani et al., 2024b;a; Sun et al., 2024; He et al., 2025b; Zheng et al., 2024; Xu et al., 2024; Watson et al., 2024; Yu et al., 2025b; Li et al., 2025; Zheng et al., 2024; He et al., 2025a; Zhou et al., 2025; Li et al., 2024) for conditioning. However, such parameter-conditioned models often generate world-inconsistent content due to the lack of explicit 3D guidance, especially in out-of-distribution scenarios. To mitigate this, recent works have shifted toward guiding generation with point-cloud renderings (anchor videos) as conditions to leverage geometric cues for more accurate camera control (Yu et al., 2024b; Popov et al., 2025; Hou & Chen, 2024; Ren et al., 2025; Zheng et al., 2025; Seo et al., 2024; Cao et al., 2025; Müller et al., 2024; Liu et al., 2024; Zhang et al., 2024a; 2025; Zhou et al., 2024; Yang et al., 2025; Bernal-Berdun et al., 2025). Alternatively, some methods rely on trajectory tracking and encoding as intermediate guidance (Jin et al., 2025; Feng et al., 2024; Xiao et al., 2024; Gu et al., 2025), but such guidance is generally less direct than anchor video conditions and often results in lower accuracy. Despite these advances, rendered anchor videos are often misaligned due to point-cloud errors, and the reliance on accurate camera estimations restricts training to static datasets. Moreover, prior methods require large-scale data to correct misalignment and increase diversity. To address these issues, we propose a masking-based anchor video construction method for precise alignment without camera annotations, and a visibility-aware ControlNet that conditions on the anchor video both efficiently and effectively.

**Video-Based Camera Control.** V2V camera control redirects camera trajectories in existing videos, with applications in filmmaking, augmented reality, and beyond. Unlike T2V and I2V, it is harder to recover comprehensive 4D information from original videos, and paired ground-truth 4D data are scarce. To overcome this, one line of work applies test-time optimization or fine-tuning on specific scenes (You et al., 2024; Zhang et al., 2024a), reducing data reliance but incurring heavy inference overhead. Another line collects large-scale paired videos from simulators such as Unreal Engine5 (Bai et al., 2025a;b), Kubric (Greff et al., 2022; Van Hoorick et al., 2024), or Animated Objaverse (Deitke et al., 2023; Wu et al., 2025; Gao et al., 2024; Yu et al., 2024a; Wang et al., 2024a), though realism and diversity remain limited. The most related works (Bian et al., 2025; Yu et al., 2025a) leverage structured 3D priors (e.g., anchor videos) for controllable V2V generation, but require extensive backbone tuning on large curated 4D datasets. By contrast, our method trains efficiently with only a small amount of I2V data and minimal backbone modification, while generalizing well to V2V.

## 3. Background: Video Diffusion Models

We build on latent video diffusion models. Given an RGB video $x \in \mathbb{R}^{L \times 3 \times H \times W}$, a pre-trained 3D-VAE encodes it into latent representations $\mathbf{z} = \mathcal{E}(x) \in \mathbb{R}^{L' \times C \times h \times w}$, where $L'$, $C$, and $h \times w$ denote the latent sequence length, channels, and spatial resolution. In the forward diffusion process, a clean latent $\mathbf{z}_0 \sim p_{\text{data}}(\mathbf{z})$ is gradually corrupted as $\mathbf{z}_t = \sqrt{\bar{\alpha}_t}\mathbf{z}_0 + \sqrt{1 - \bar{\alpha}_t}\boldsymbol{\epsilon}$, with $\boldsymbol{\epsilon} \sim \mathcal{N}(0, I)$. The model learns to predict $\boldsymbol{\epsilon}$ from $\mathbf{z}_t$ conditioned on external signals $c$ (e.g., image or text) by minimizing $\mathcal{L}_{\text{denoise}} = \mathbb{E}_{\mathbf{z}_0, t, \boldsymbol{\epsilon}, c}[\|\boldsymbol{\epsilon}_\theta(\mathbf{z}_t, t, c) - \boldsymbol{\epsilon}\|_2^2]$. During inference, the model denoises from Gaussian noise to obtain $\hat{\mathbf{z}}$, which is decoded by the VAE decoder $\mathcal{D}$ to output video $\hat{\mathbf{x}} = \mathcal{D}(\hat{\mathbf{z}})$.

**Base Model.** We adopt CogVideoX-5B-I2V (Yang et al., 2024b) which supports both image and text conditions as our base model. It employs a DiT-style (Peebles & Xie, 2023) backbone with full 3D self-attention to jointly model spatial and temporal dependencies across video frames.

**Guiding VDMs with Anchor Video as a Structured Prior for Camera Control.** Recent methods (Yu et al., 2024b; 2025a; Cao et al., 2025; Zhang et al., 2024a) leverage *anchor videos* to enable controllable video generation with explicit camera motion. These anchors are typically rendered by lifting a single RGB image into 3D point clouds (Wang et al., 2024b; Yang et al., 2024a) and re-rendering it along a camera trajectory, providing structured geometry and motion priors to guide generation. During training, anchor videos are rendered from the first frame of the source video along its original camera trajectory, and the model learns to reconstruct the video conditioned on this anchor. At inference, anchors are similarly generated from an input image and a user-defined trajectory.

However, existing approaches suffer from two key limitations. First, anchor videos based on imperfect 3D reconstructions are often inaccurate, forcing the model to both inpaint missing regions and correct misaligned visible areas, leading to inefficient learning (Fig. 5 (a)). Second, latent-space anchor conditioning usually requires fine-tuning the

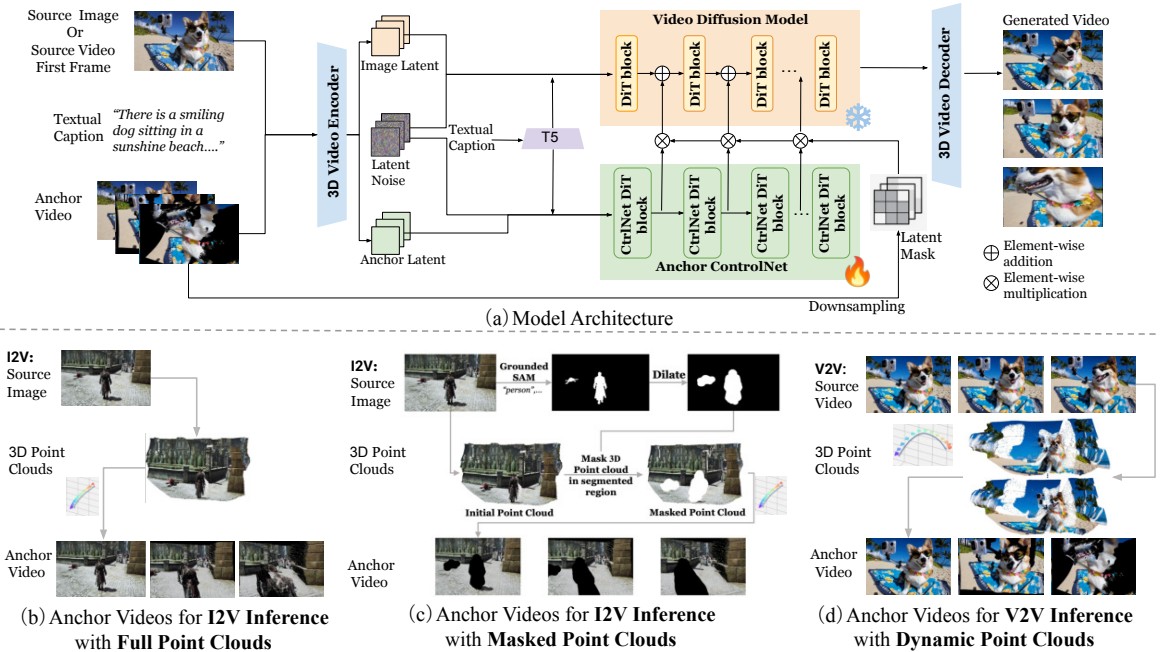

(a) Model Architecture

(b) Anchor Videos for **I2V Inference** with **Full Point Clouds**

(c) Anchor Videos for **I2V Inference** with **Masked Point Clouds**

(d) Anchor Videos for **V2V Inference** with **Dynamic Point Clouds**

*Figure 2.* EPiC Model Architecture. (a): Overview of EPiC framework. EPiC supports multiple inference scenarios. (b) and (c) illustrate our I2V inference scenarios using full and masked point clouds. (d): V2V inference scenario employing dynamic point clouds.

backbone or injecting heavy modules, increasing computation and hurting generalization (Tab. 1). To address these issues, we propose EPiC, an efficient framework that learns precise camera control using masking-based anchor videos and a lightweight Anchor-ControlNet, as detailed below.

## 4. EPiC: An Efficient Framework for Camera Control Learning

The overall framework is illustrated in Fig. 2. We first construct precisely aligned anchor and source videos as training input-output pairs with a visibility-based masking strategy (Sec. 4.1). Then, we introduce a lightweight Anchor-ControlNet that learns to reconstruct the source video from the anchor video efficiently (Sec. 4.2). Finally, we describe our training and inference details (Sec. 4.3).

### 4.1. Constructing Precise Anchor Videos from Source Videos via Visibility-Based Masking

We aim to construct anchor videos that are well-aligned with the source videos, making the learning process easier and more efficient. To achieve this, we use the following two steps to construct anchor videos through a masking strategy that preserves alignment while mimicking the geometric characteristics of point-cloud-rendered videos:

**Step 1: Pixel-Level Visibility Tracking and Masking.** We estimate pixel trajectories in the source video using dense optical flow from the first frame (computed via

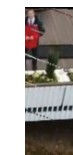

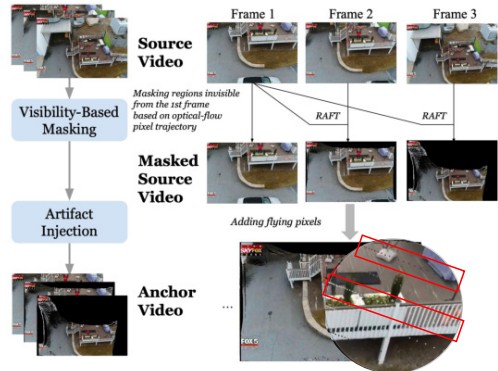

*Figure 3.* Anchor video construction.

RAFT (Teed & Deng, 2020)) to determine whether each pixel remains visible from the original viewpoint. This pixel tracking simulates how content moves or disappears due to viewpoint shifts or occlusion. We provide a binary visibility mask for each frame based on such tracking information, retaining only regions consistently traced from the original view and masking out the rest. This process effectively mimics the core property of anchor videos, which excludes newly revealed content while ensuring precise alignment in the visible regions. In cases where the visible region becomes too small due to large viewpoint shifts, we freeze the mask in subsequent frames to prevent further degradation. The masked source video is obtained by applying the visibility mask to the source video, as shown in Fig. 3.

**Step 2: Artifact Injection.** A major limitation of esti-

mated point clouds is the presence of flying-pixel artifacts, especially around object boundaries (see Fig.2(d), where splatted flying pixels appear near the dog's edges in both point cloud examples). These errors propagate to the anchor video, resulting in flying-pixel artifacts (see Fig.2(d)). To improve robustness, we simulate this flying-pixel effect during training by injecting synthetic dashed rays into the masked anchor video to better align training and inference gap (see Fig. 3 bottom red box). Specifically, we randomly sample a direction and draw multiple rays perpendicular to it, with colors sampled from the first frame to ensure temporal consistency. These rays are faded and dashed to resemble flying-pixel artifacts, and are applied only within the visible regions defined by the mask, which helps the model learn to ignore such artifacts during inference. The final artifact-injected anchor video is used for training.

### 4.2. Guiding Video Diffusion with Anchor-ControlNet

We introduce Anchor-ControlNet to guide video diffusion model with the constructed anchor video as condition (Fig. 2 (a)). We use minimal parameters for downstream adaptation to preserve the model's core generation capability (Ruiz et al., 2023) instead of full fine-tuning. To this end, we adopt a lightweight ControlNet design (<30M parameters) and keep the entire backbone frozen during training.

**Model Architecture.** Anchor-ControlNet is a lightweight DiT-based module designed to inject anchor video guidance into the base diffusion model. Given an anchor video $\mathbf{A}$, we encode it using the 3D VAE from the backbone model to obtain latent features $\mathbf{z}_{\text{anchor}}$. During the reverse diffusion process, the noisy latent $\mathbf{z}_t$ is concatenated with $\mathbf{z}_{\text{anchor}}$ along the channel dimension. The combined representation is then patchified and fed into the ControlNet DiT block. The DiT block in Anchor-ControlNet adopts a reduced hidden dimension (256 compared to 3072 in the base model) to maintain efficiency. Its output is projected back to match the backbone's dimension and added to the corresponding layer in the base DiT model. The projection layer is zero-initialized, following the standard practice in ControlNet, to ensure stable integration at the beginning of training.

**Visibility-Aware Output Masking.** Previous work, such as ViewCrafter (Yu et al., 2024b), condition directly on the entire anchor video without visibility awareness. This forces the model to simultaneously repair misaligned regions and inpaint invisible (black) areas, making the learning task unnecessarily difficult and increasing the risk of incorrect region repair during inference (In fact, we also found that simply conditioning on the entire anchor video with ControlNet makes it difficult for the model to learn invisible-region completion, causing it to follow errors present in those invisible areas (Fig. 5 (c))). TrajectoryCrafter (Yu et al., 2025a) incorporates visibility information by encoding the visibil-

ity mask into latents, which forces the model to learn the complex relationship among the anchor video, source video, and the mask, thereby increasing training difficulty.

In contrast, we address these issues by manually distinguishing visible and invisible content: the ControlNet focuses solely on copying visible content, while the synthesis of occluded or invisible regions is entirely delegated to the base diffusion model. Formally, we require the control signal from the anchor video to only affect visible regions by applying a binary mask $M \in \{0, 1\}^{T' \times h \times w}$ to the ControlNet output. The mask is downsampled to match the latent resolution and used to update the base model's latent features (Fig. 2a). ControlNet output is computed as $\tilde{\mathbf{z}} = \text{Proj}(\text{DiT}_{\text{ctrl}}([\mathbf{z}_t, \mathbf{z}_{\text{anchor}}]))$ and fused with the base model: $\hat{\mathbf{z}} = \text{DiT}_{\text{base}}(\mathbf{z}_t) + M \odot \tilde{\mathbf{z}}$, where $M$ masks out invisible regions. This visibility-aware latent fusion is applied during both training and inference, allowing the base model to inpaint disoccluded regions while Anchor-ControlNet controls the visible content aligned with the anchor video.

### 4.3. Training and Inference

**Training.** We create our masking-based anchor video from in-the-wild source videos to construct training data. We train the Anchor-ControlNet on our collected anchor and source video pairs by conditioning on the anchor video to predict the source video with the standard diffusion Loss. Details of our in-the-wild video data are provided in Sec. 5.1.

**I2V Inference.** We consider two distinct inference scenarios for I2V: mode (b): **with full point clouds** (illustrated in Fig. 2 (b)) and mode (c) **with masked point clouds** (shown in Fig. 2 (c)). In the first scenario, given an input image and a target camera trajectory, we first estimate the metric depth using DAv2 (Yang et al., 2024a), then unproject the image into a 3D point cloud and render the anchor video along the specified camera trajectory. However, this approach produces anchor videos where objects remain static, as rendering is performed from a stationary point cloud. To overcome this limitation and support **dynamic object movement** while preserving precise camera control, we propose inference with masked point clouds. Specifically, given a single input image, we use GroundedSAM (Ren et al., 2024) to identify and segment potentially dynamic objects (*e.g.*, "person", "animal") from a predefined category list. Users may also customize tailored segmentation masks. During 3D point cloud projection, we exclude points within the segmented regions. These masked areas are omitted when rendering the anchor video, which allows the reserved background to drive camera motion while leaving the segmented foreground objects unconstrained, enabling natural movement within the generated video.

**V2V Inference.** EPiC also supports V2V camera control (Fig. 2 (d)). Given an input video, we apply

*Table 1.* Quantitative evaluation results on RealEstate10K (Zhou et al., 2018b) and MiraData (Ju et al., 2024) for I2V camera control. The best numbers are in **bold**. The Total score is the average of all quality metrics. † indicates re-implementation results on I2V.

| Dataset | Method | | Quality Score | | | | | | Camera Score | | |
| --- | --- | --- | --- | --- | --- | --- | --- | --- | --- | --- | --- |
| | | Total | Subject Consist | Bg Consist | Motion Smooth | Temporal Flicker | Aesthetic Quality | Imaging Quality | Rotation Error (↓) | Transition Error (↓) | CamMC (↓) |
| RE10K | CameraCtrl (He et al., 2024) | 78.35 | 89.95 | 91.25 | 97.16 | 91.99 | 43.32 | 56.43 | $1.12 \pm 0.44$ | $1.78 \pm 0.93$ | $2.36 \pm 1.01$ |
| | AC3D† (Bahmani et al., 2024a) | 82.63 | **91.96** | 92.77 | 98.30 | 96.23 | 50.97 | **65.56** | $0.86 \pm 0.37$ | $1.50 \pm 0.82$ | $1.97 \pm 0.86$ |
| | ViewCrafter (Yu et al., 2024b) | 81.18 | 90.23 | 92.99 | 97.74 | 93.51 | 48.29 | 64.33 | $0.50 \pm 0.16$ | $1.05 \pm 0.32$ | $1.35 \pm 0.40$ |
| | FloVD (Jin et al., 2025) | 82.61 | 91.77 | 93.25 | 98.30 | 96.23 | 50.97 | 65.16 | $0.76 \pm 0.31$ | $1.14 \pm 0.52$ | $1.47 \pm 0.56$ |
| | Gen3C (Ren et al., 2025) | 82.27 | 91.10 | 92.75 | 97.99 | **96.67** | 50.61 | 64.54 | $0.45 \pm 0.13$ | $0.99 \pm 0.22$ | $1.35 \pm 0.30$ |
| | EPiC (Ours) | **82.63** | 91.62 | **93.43** | **98.48** | 96.47 | **51.19** | 64.57 | $\mathbf{0.40} \pm 0.11$ | $\mathbf{0.86} \pm 0.18$ | $\mathbf{1.17} \pm 0.23$ |
| MIRA | CameraCtrl (He et al., 2024) | 78.06 | 89.28 | 91.15 | 97.30 | 90.22 | 49.35 | 51.11 | $1.62 \pm 0.84$ | $4.67 \pm 1.47$ | $5.66 \pm 2.06$ |
| | AC3D† (Bahmani et al., 2024a) | 82.78 | 91.75 | 92.81 | 98.20 | 94.77 | 57.64 | **61.51** | $1.13 \pm 0.74$ | $3.98 \pm 1.50$ | $4.79 \pm 1.53$ |
| | ViewCrafter (Yu et al., 2024b) | 79.87 | 86.56 | 91.55 | 96.26 | 91.71 | 54.21 | 58.92 | $1.16 \pm 0.34$ | $2.95 \pm 0.98$ | $3.42 \pm 1.04$ |
| | FloVD (Jin et al., 2025) | 82.55 | 91.64 | 92.91 | 98.43 | 94.67 | 57.46 | 60.21 | $0.95 \pm 0.44$ | $2.15 \pm 0.98$ | $3.48 \pm 1.03$ |
| | Gen3C (Ren et al., 2025) | 80.50 | 88.56 | 90.75 | 96.76 | 91.74 | 55.21 | 59.98 | $0.81 \pm 0.24$ | $2.05 \pm 0.77$ | $2.75 \pm 0.72$ |
| | EPiC (Ours) | **82.89** | **91.82** | **92.94** | **98.75** | **94.86** | **57.94** | 61.03 | $\mathbf{0.66} \pm 0.22$ | $\mathbf{1.78} \pm 0.67$ | $\mathbf{2.10} \pm 0.60$ |

DepthCrafter (Hu et al., 2024) to estimate continuous depths and construct dynamic point clouds. The anchor video is rendered by replaying the target trajectory over 4D representation. Note that because DepthCrafter predicts depth in each frame's camera coordinate, the reconstructed 4D point cloud is also camera-centric, rather than defined in a global frame. Therefore, the applied trajectory is interpreted as a relative transformation on top of the source motion. Additionally, since the base I2V model is frozen, we provide the first frame of the conditional video as input to the model.

## 5. Experiments

### 5.1. Experimental Setup

**Datasets and Baselines.** We compare EPiC and recent baselines for I2V setting on the RealCam-Vid test set (Li et al., 2025) from two data source, RealEstate10K (RE10K) (Zhou et al., 2018b) and MiraData (MIRA) (Ju et al., 2024), consisting of both static and dynamic scenes. We sample 500 videos for each dataset. For baselines, we consider SoTA methods including CameraCtrl (He et al., 2024), AC3D (Bahmani et al., 2024a), ViewCrafter (Yu et al., 2024b), FloVD (Jin et al., 2025), and Gen3C (Ren et al., 2025). For consistency, we use similar anchor videos per test sample for both ViewCrafter and EPiC. For the V2V setting, we qualitatively evaluate using Sora videos (Brooks et al., 2024) and challenging movie clips, while providing quantitative results on sampled 100 Kubric4D (Greff et al., 2022) scenes. We use GCD (Van Hoorick et al., 2024), TrajectoryCrafter (Yu et al., 2025a), ReCamMaster (Bai et al., 2025a), and Gen3C (Ren et al., 2025) as V2V baselines.

**Implementation Details.** EPiC is trained on 5,000 videos from Panda70M dataset (Chen et al., 2024) for 500 iterations, using a batch size of 16 across 8 40G A100 GPUs. The text condition for the I2V backbone is obtained from the annotated captions in Panda70M. Training takes less than 3 hours with a learning rate of $2 \times 10^{-4}$ with

AdamW (Loshchilov, 2017) optimizer. During inference, we use classifier-free guidance (CFG) with a scale of 6.0 for text conditioning. More details are in the Appendix Sec. B.1.

**Evaluation Metrics.** For camera-related metrics, we follow prior works (Wang et al., 2024d; He et al., 2024) and report Rotation Error (RotError), Translation Error (TransError), and CamMC, which respectively measure orientation differences, positional errors, and overall camera pose consistency between the predicted and ground-truth trajectories. To account for randomness, we sample five fixed random seeds per test instance and report the mean and standard deviation of each camera metric. For visual quality, we adopt VBench (Huang et al., 2024) metrics including Subject Consistency, Background Consistency, Motion Smoothness, Temporal Flickering, Aesthetic Quality, and Imaging Quality. Metrics details are provided in Appendix B.2.

### 5.2. Quantitative Evaluation

**Performance.** In Tab. 1, we compare EPiC and recent SOTA I2V camera control methods on RE10K and MIRA. EPiC achieves comparable quality scores to those of prior approaches across both the RE10K and MIRA benchmarks. EPiC attains the highest total score on both datasets, suggesting strong subject/background consistency, smooth motion, and reduced temporal flicker. Furthermore, our method significantly outperforms existing baselines in all three camera score metrics. This demonstrates superior fidelity in controlling camera motions with the best robustness across seeds, as reflected by the lowest standard deviations.

For V2V camera control, results on Kubric-4D (Tab. 2) show that our method, although only trained on I2V data, is comparable with strong baselines specifically trained for this task such as GCD and TrajCrafter, demonstrating its strong zero-shot generalization ability.

**Efficiency.** In Fig. 1 and Appendix Tab. 4, we present a comparison of training efficiency for I2V and V2V. EPiC

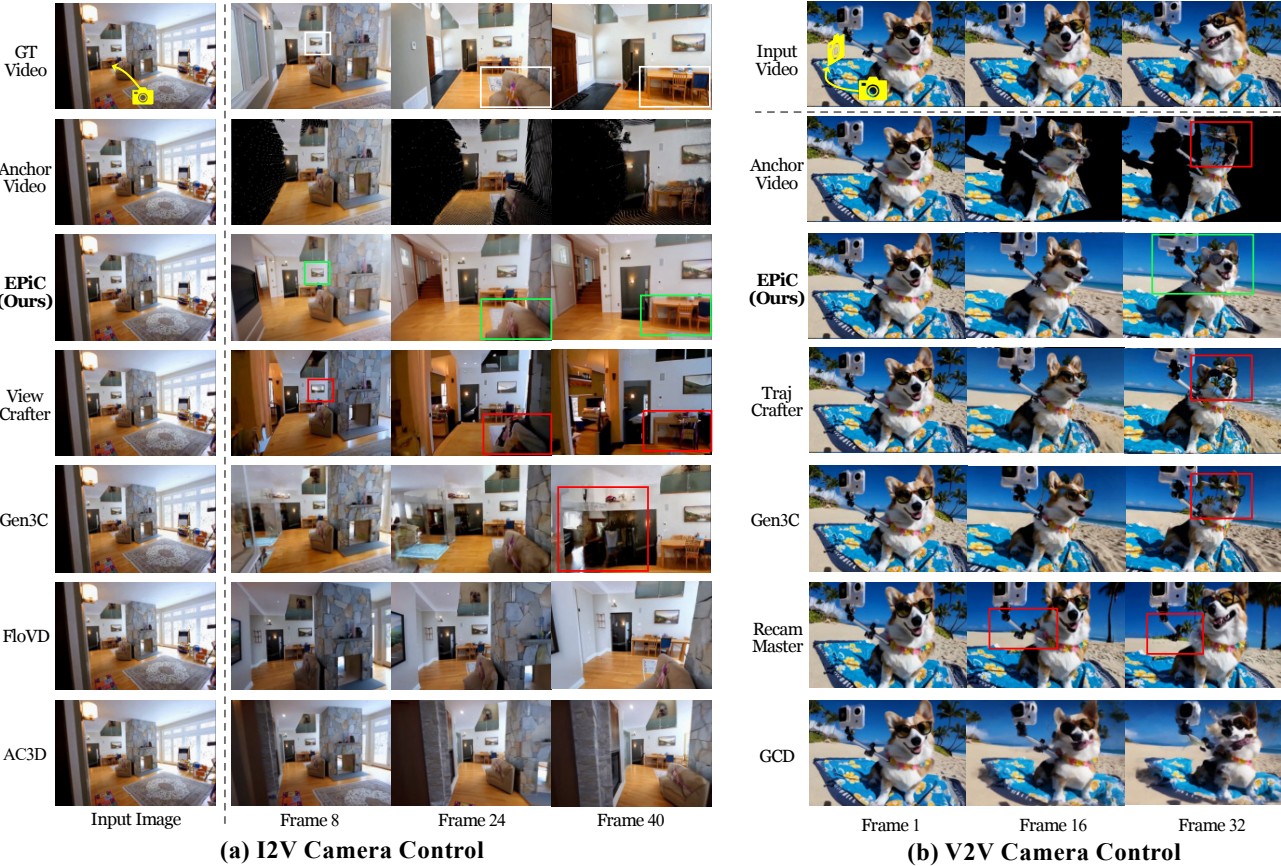

**(a) I2V Camera Control**

**(b) V2V Camera Control**

*Figure 4.* Generated videos comparing with other camera control methods for I2V and V2V tasks.

requires over an order of magnitude fewer training data and substantially lower training cost, while also using significantly fewer parameters, requiring only 15 GPU hours to train. Importantly, quantitative results show that our method achieves comparable or even superior performance, showing that accurate and robust camera control capability can be achieved without relying on heavy data or computation.

### 5.3. Qualitative Examples

Fig. 4 compares camera control results from EPiC and SOTA open-source baselines on both I2V and V2V settings. For I2V, we include ViewCrafter, AC3D, FloVD and Gen3C; for V2V, we compare against GCD, TrajectoryCrafter, Gen3C and ReCamMaster. AC3D, GCD, and ReCamMaster condition on camera embeddings, while ViewCrafter, TrajectoryCrafter, and Gen3C, like ours, condition on anchor videos. FloVD instead uses optical-flow maps as its control signal.

**I2V Camera Control.** As shown in Fig. 4(a), ViewCrafter (4th row), Gen3C (5th row), and our method (3rd row) can follow anchor videos. However, ViewCrafter often introduces content inconsistencies (red boxes), such as gradually changing a painting into a glass-like material (2nd

column) and producing severe distortions around the sofa (3rd column) and chairs (4th column). This is likely due to over-repairing misaligned regions when trained with point-cloud-based anchor videos. Gen3C struggles under large camera motion and generates messy content in invisible regions (4th column). In contrast, our method preserves visible content thanks to aligned anchor supervision (green boxes) and generates reasonable content in invisible regions. Methods without anchor guidance (AC3D and FloVD) fail to follow the camera trajectory. Notably, this sample is from the Real10K (in-domain for ViewCrafter, AC3D, and Gen3C), yet EPiC achieves better accuracy and visual quality. Additional qualitative results are provided in Appendix Figs. 12 to 14, and more in-the-wild examples in Fig. 17.

**V2V Camera Control.** Fig. 4(b) shows V2V camera control results. GCD produces blurry foregrounds and low-fidelity details, while TrajCrafter, Gen3C, and our method can generally follow anchor videos. However, incorrect occlusion appears in the 3rd frame of the anchor video, where the tree passes through regions missing in the reconstructed point cloud. TrajCrafter and Gen3C directly follow this erroneous signal (red box), likely due to their heavily modified backbones that enforce anchor adherence even when the renderer

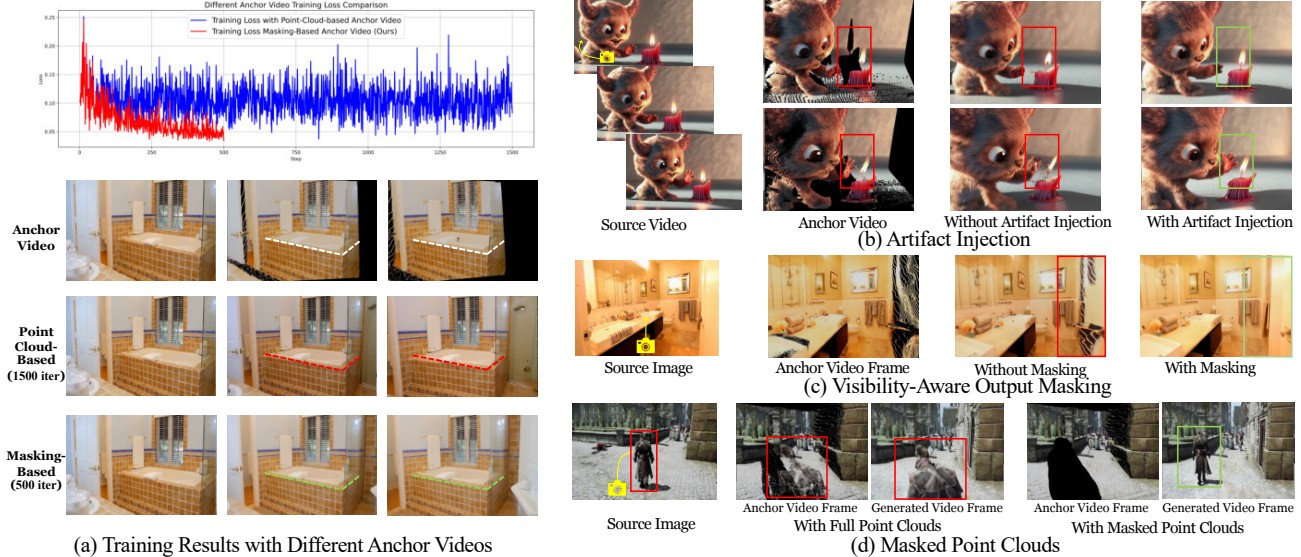

*Figure 5.* Qualitative examples for ablation study.

is inaccurate. In contrast, our method freezes the backbone and uses the anchor video only as guidance, allowing the model to generate the most plausible content and avoid being misled by incorrect occlusions (green box). Also, ReCamMaster fails to preserve the selfie stick, while EPiC successfully maintains it thanks to explicit 3D guidance from the anchor video. We provide more qualitative comparisons and in-the-wild examples in Figs. 15, 16, 18 and 19, as well as multi-camera shooting examples in Fig. 20.

### 5.4. Ablation Studies

In this section, we present ablation studies for key components of our framework. We analyze the impact of different anchor video constructions, artifact injection, visibility-aware output masking, and masked point clouds for dynamic objects. We also provide experiments on the effects of training data sources, lightweight model design, generalization to different backbones, and more detailed ablations on Anchor-ControlNet's visibility-aware masking in Appendix D.

**Effects of Different Types of Anchor Videos.** We compare the effect of three types of anchor video training data on camera control performance in Tab. 3 and Fig. 5(a): point cloud-based, mixed (50% point cloud-based + 50% masking-based), and fully masking-based, using 5K RealEstate10K videos with large camera motion. As shown in Tab. 3, point cloud anchors yield the highest camera errors and variance despite using 3× more training iterations, while masking-based anchors achieve the best results across all metrics. Due to misalignment, point cloud anchors also converge much slower and incur significantly higher loss (Fig. 5(a)). Qualitatively, point cloud anchors lead to misaligned geometry (red dashed lines), while ours faithfully follows the

anchor (green dashed lines). To further understand this gap, we quantify anchor quality by measuring the anchor-source PSNR in visible regions (Tab. 3). From point cloud-based to mixed to masking-based anchors, the anchor PSNR increases monotonically, and all camera metrics consistently improve accordingly, confirming that anchor alignment quality is the key factor to downstream camera control accuracy.

**Effects of Artifact Injection for Constructing Training Anchor Videos.** Fig. 5 (b) shows the effectiveness of artifact injection, as described in Sec. 4.1. Due to point cloud estimation errors, flying pixels often appear when rendering from rapidly changing camera poses, resulting in incorrect guidance even within visible regions. Without artifact injection, the model follows these flawed inputs, leading to similar artifacts at inference (red box). In contrast, with artifact injection, the model learns to repair such artifacts during training, resulting in cleaner outputs (green box).

**Effects of Visibility-Aware Output Masking.** One crucial design in our Anchor-ControlNet is the visibility-aware output masking strategy, which enables the model to control only the visible regions, as described in Sec. 4.2. We conduct an ablation study by training modules without mask awareness, similar to ViewCrafter. As shown in Fig. 5 (c), without output masking, the model is influenced by tearing artifacts rendered from the point cloud, which guide it to generate ambiguous content in these corrupted regions (see red boxes). In contrast, our method excludes such regions from the control signal, allowing the model to generate reasonable and faithful content (green boxes).

**Effects of Masked Point Clouds for Dynamic Objects.** Fig. 5 (d) shows examples of results using the masked point cloud to enable dynamic objects, as described in Sec. 4.3.

*Table 2.* V2V results on Kubric-4D.

| Method | PSNR ↑ | SSIM ↑ |
|---|---|---|
| GCD (Van Hoorick et al., 2024) | 19.72 | 0.59 |
| TrajCrafter (Yu et al., 2025a) | 19.61 | 0.62 |
| Gen3C (Ren et al., 2025) | 19.69 | 0.61 |
| EPiC (Ours) | 19.65 | 0.60 |

*Table 3.* Ablation on anchor video type on RE10K. Anchor PSNR measures pixel-level alignment between anchors and source videos in visible regions.

| Anchor Video Type | Anchor PSNR (↑) | RotErr (↓) | TransErr (↓) | CamMC (↓) |
|---|---|---|---|---|
| Point cloud-based (1500 iters) | 16.01 | $0.60 \pm 0.20$ | $1.07 \pm 0.39$ | $1.45 \pm 0.62$ |
| Mixed (50% PC + 50% Mask, 1000 iters) | 28.07 | $0.48 \pm 0.15$ | $0.95 \pm 0.28$ | $1.29 \pm 0.40$ |
| Masking-based (500 iters; Ours) | **40.12** | **0.40** $\pm 0.11$ | **0.86** $\pm 0.18$ | **1.17** $\pm 0.23$ |

Without masking (with full point cloud, mode (b) in Fig. 2), the generated video is static—the character (in the red boxes) stands still due to strong 3D guidance in the anchor video. In contrast, masking the point cloud (mode (c) in Fig. 2) removes control signals from the character, allowing it to move freely and enabling a natural walking motion (as shown in the green box). Appendix Fig. 21 contains more examples showing our dynamic object control ability.

## 6. Conclusion

We propose EPiC, an efficient framework for precise camera control. It constructs high-quality training anchors by masking source videos using first-frame visibility, eliminating the need for camera pose estimation and enabling robust application to in-the-wild videos. We further introduce Anchor-ControlNet, a lightweight adapter that copies visible regions from anchors without modifying the backbone, achieving superior visual quality and camera accuracy over prior methods on I2V and V2V tasks.

## Acknowledgments

This work was supported by DARPA ECOLE Program No. HR00112390060, NSF-AI Engage Institute DRL-2112635, DARPA Machine Commonsense (MCS) Grant N66001-19-2-4031, ARO Award W911NF2110220, ONR Grant N00014-23-1-2356, Accelerate Foundation Models Research program, and a Bloomberg Data Science PhD Fellowship. The views contained in this article are those of the authors and not of the funding agency.

## Impact Statement

This work studies controllable video generation with precise camera motion control, aiming to improve the reliability and usability of generative models in visual content creation. The proposed method can benefit applications in film production, virtual environment creation, robotics simulation, and data augmentation. As with other generative video models, our approach may be misused to generate misleading visual content. However, our method introduces no new capabilities beyond existing video generation systems, and focuses on improving controllability rather than realism of identity or sensitive attributes. We encourage responsible use and adherence to existing ethical guidelines.

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

## A. Anchor Video Constructing Method Illustration

We provide an illustration of anchor video construction in Figure 6. (a) Previous methods rely on lifting the first frame into a 3D point cloud and rendering along estimated camera trajectories. This often leads to misaligned visible regions due to pose/depth estimation errors, requiring large-scale datasets and many training iterations. (b) In contrast, our visibility-based masking approach directly preserves only pixels that can be traced back to the first frame, producing well-aligned anchor videos without any camera pose estimation. This design greatly simplifies learning and enables efficient training with substantially fewer videos and iterations.

## B. Experiment Details

### B.1. Implementation Details

EPiC is trained on a subset of $5,000$ videos from the Panda70M dataset (Chen et al., 2024) for 500 iterations, using a total batch size of 16 across 8 40GB A100 GPUs. The text condition for the I2V backbone is obtained from the annotated captions in Panda70M. The subset is selected based on optical flow scores, where we rank videos by their average flow magnitude and retain those with sufficient motion to ensure meaningful camera control training. Training takes less than 3 hours with a learning rate of $2 \times 10^{-4}$, using the AdamW (Loshchilov, 2017) optimizer. For our visibility-aware output masking, we apply average pooling to downsample the raw visibility mask to the latent resolution. We train the Anchor-ControlNet at a resolution of $480 \times 720$ for 49 frames per video (which is the default setting of CogVideoX-5B-I2V (Yang et al., 2024b)), with ControlNet weights set to 1.0.

During inference, we apply classifier-free guidance (CFG) (Ho & Salimans, 2022) with a scale of 6.0 for text conditioning. Following AC3D (Bahmani et al., 2024a), we only inject the ControlNet into the first 40% diffusion steps at inference. We apply max pooling to downsample the raw visibility mask to the latent resolution for visibility-aware output masking. For videos with caption annotations, we directly use the annotations as the textual condition. For those without annotations, we either generate the text condition using advanced vision-language models (Li et al., 2023; Bai et al., 2023) based on the visual input, or manually write prompts for specific usage scenarios.

### B.2. Evaluation Metrics

We adopt three standard camera pose evaluation metrics to measure the alignment between predicted and ground-truth camera trajectories: **Rotation Error (RotErr)**, **Translation Error (TransErr)**, and **Camera Matrix Consistency (CamMC)** following MotionCtrl (Wang et al., 2024d) and CameraCtrl (He et al., 2024).

- **Rotation Error (RotErr)** measures the angular deviation (in radians) between the predicted and ground-truth camera rotations:

$$\text{RotErr} = \sum_{i=1}^{n} \arccos\left(\frac{\text{tr}(\tilde{R}_i R_i^\top) - 1}{2}\right)$$

  where $\tilde{R}_i$ and $R_i$ are the predicted and ground-truth rotation matrices at frame $i$, and $n$ is the number of frames in the video.
- **Translation Error (TransErr)** computes the $\mathcal{L}_2$ distance between normalized translation vectors:

$$\text{TransErr} = \sum_{i=1}^{n} \left\| \frac{\tilde{T}_i}{\tilde{s}_i} - \frac{T_i}{s_i} \right\|_2$$

  where $\tilde{T}_i$ and $T_i$ are the predicted and ground-truth camera translations, and $\tilde{s}_i$, $s_i$ are their respective scene scales—defined as the $\mathcal{L}_2$ distance between the first and farthest frame in each video.
- **Camera Matrix Consistency (CamMC)** evaluates overall pose alignment by comparing full camera-to-world matrices with scale normalization:

$$\text{CamMC} = \sum_{i=1}^{n} \left\| \left[ \tilde{R}_i \ \frac{\tilde{T}_i}{\tilde{s}_i} \right]^{3\times4} - \left[ R_i \ \frac{T_i}{s_i} \right]^{3\times4} \right\|_2$$

  where $\tilde{R}_i$, $\tilde{T}_i$, and $\tilde{s}_i$ are the predicted rotation, translation, and scene scale; $R_i$, $T_i$, and $s_i$ are their ground-truth counterparts.

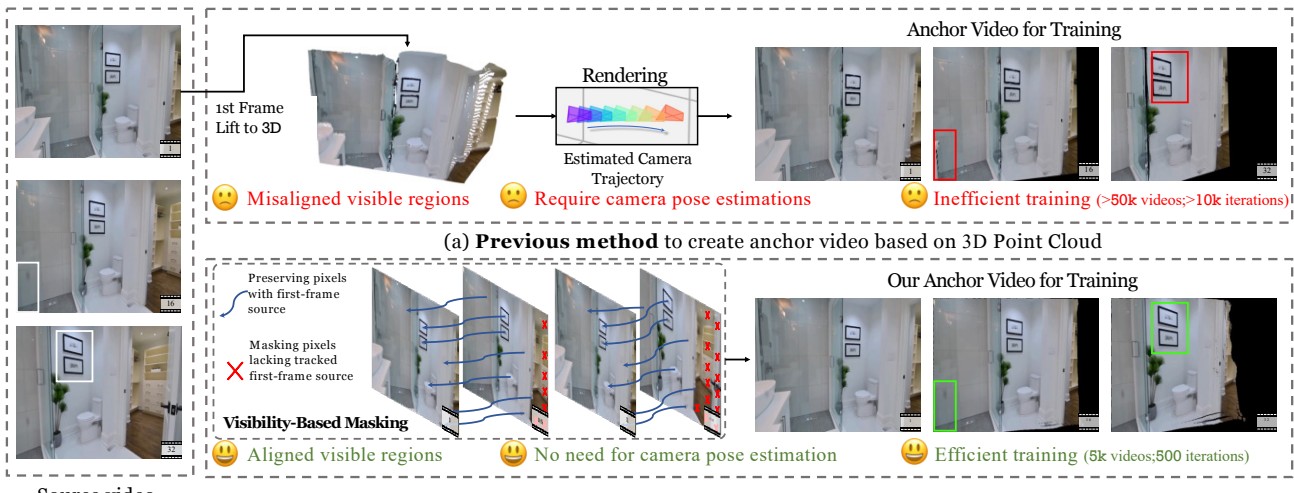

*Figure 6.* Comparison between prior 3D point cloud–based anchor video construction and our visibility-based masking approach.

*Table 4.* Efficiency comparison across methods. 'Steps' denotes the number of training iterations, and '#Videos' denotes the amount of training data.

| Method | Steps | Batch Size | Steps×Batch Size | #Videos | #Parameters |
|---|---|---|---|---|---|
| TrajCrafter (Yu et al., 2025a) | 150k | 8 | 1200k | 632k | 5.57B |
| ViewCrafter (Yu et al., 2024b) | 40k | 18 | 720k | 632k | 1.44B |
| AC3D (Bahmani et al., 2024a) | 750k | 32 | 24000k | 100k | 200M |
| CameraCtrl (He et al., 2025a) | 50k | 32 | 1600k | 65k | 211M |
| GCD (Van Hoorick et al., 2024) | 10k | 56 | 560k | 77k | 2.41B |
| Gen3C (Ren et al., 2025) | 10k | 64 | 640k | 100k | 7.23B |
| FloVD (Jin et al., 2025) | 50k | 16 | 800k | 600k | 1.40B |
| ReCamMaster (Bai et al., 2025a) | 20k | 8 | 160k | 136k | 1.49B |
| EPiC (Ours) | 0.5k | 8 | **4k** | **5k** | **26M** |

For visual quality, we adopt the evaluation protocol from VBench (Huang et al., 2024), including metrics such as Subject Consistency, Background Consistency, Motion Smoothness, Temporal Flickering, Aesthetic Quality, and Imaging Quality. We refer to VBench (Huang et al., 2024) for more details.

## C. Full Efficiency Comparison

We provide full efficiency comparison in Table 4. As shown, EPiC achieves over an order-of-magnitude improvement in compute cost, training data size, and parameter efficiency.

## D. Additional Experiments

In this section, we provide additional ablations on the training data, the use of Anchor-ControlNet, and the lightweight ControlNet design.

### D.1. Effects of Training Data Sources

A key advantage of our method is that it does not rely on camera pose annotations, which enables training on diverse, in-the-wild video datasets beyond multi-view datasets with limited domain coverage. To validate this, we conduct an ablation comparing training on the widely used RealEstate10K (Zhou et al., 2018b), which is a mulit-view dataset limited to static indoor scenes, with training on Panda70M (Chen et al., 2024), which contains more diverse and dynamic videos.

We report quantitative results in Tab. 5. We observe that both data sources yield comparable performance on RealEstate10K,

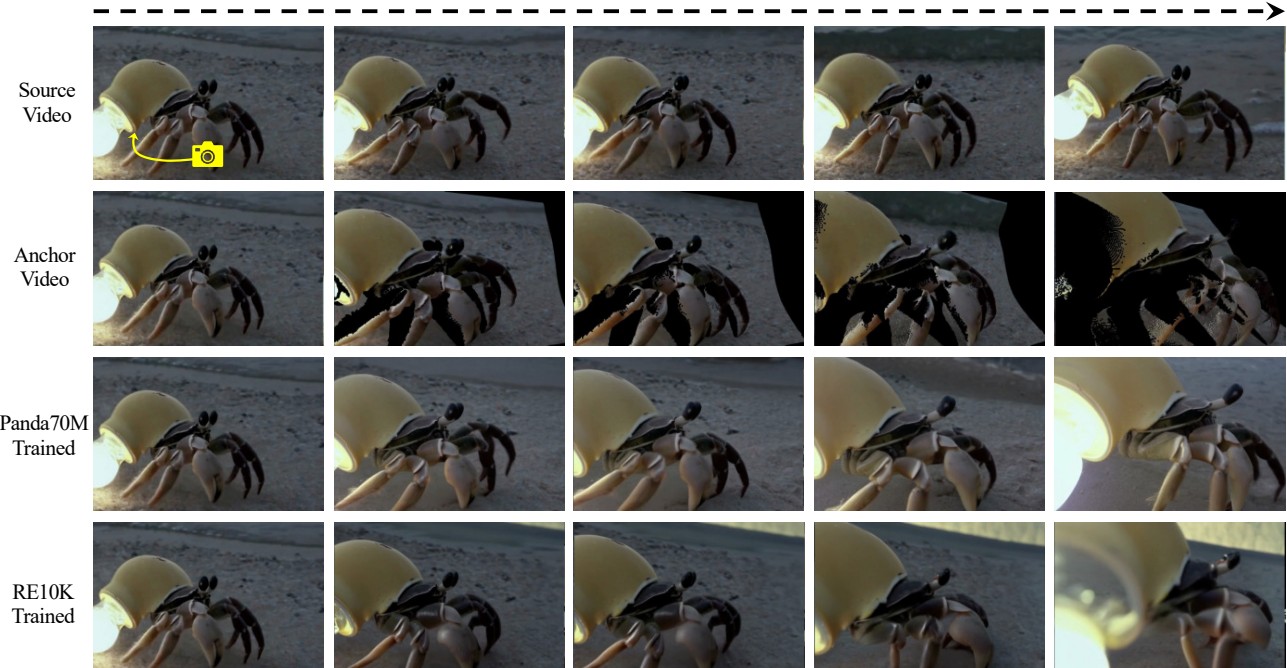

*Figure 7.* Qualitative V2V camera control results of models trained from different data sources.

*Table 5.* Ablation of using different data sources for training EPiC.

| Training Data Source | RealEstate10K | | | MiraData | | |
|---|---|---|---|---|---|---|
| | Rot. Err (↓) | Trans. Err (↓) | CamMC (↓) | Rot. Err (↓) | Trans. Err (↓) | CamMC (↓) |
| RealEstate10K (Zhou et al., 2018b) | 0.43 ±0.10 | 0.84 ±0.22 | 1.06 ±0.25 | 0.73 ±0.32 | 1.88 ±0.75 | 2.21 ±0.65 |
| Panda70M (Chen et al., 2024) | 0.40 ±0.11 | 0.86 ±0.18 | 1.17 ±0.23 | 0.66 ±0.22 | 1.78 ±0.67 | 2.10 ±0.60 |

while training with Panda70M achieves slightly better results on MiraData, likely due to its more diverse training content. However, in the V2V setting, especially when the reference video involves fine-grained motion (*e.g.*, detailed limb articulation), models trained on RealEstate10K fail to generalize effectively. Specifically, as shown in Fig. 7, the crab's legs exhibit intricate, localized motion patterns. While the model trained on Panda70M is able to precisely follow these details by following the anchor video, the model trained on RealEstate10K can only capture a coarse moving direction, failing to reproduce the fine motion in the crab's legs. This limitation is likely due to the lack of diverse and dynamic videos in the RealEstate10K dataset, which mainly consists of indoor scenes that differ significantly from the domain of the crab video.

### D.2. Effects of Lightweight Anchor-ControlNet Design

We ablate the design of our lightweight ControlNet in Tab. 7. Specifically, we compare injecting into half of the backbone layers (21 layers here (CogVideoX-5B-I2V has 42 layers totally), as in the default ControlNet setting) with and without using pretrained weights, and further study the effect of reducing the number of injection layers. Our results show that using a high-dimensional feature space (3072) with pretrained CogVideoX weights performs comparably to using no pretraining and a much smaller dimension (256), suggesting that the region-copying control is relatively easy to learn. In addition, reducing the number of injection layers to 8 does not hurt performance, while further reducing it to only 2 layers results in a noticeable decreased control accuracy. Based on these findings, we adopt the most cost-effective configuration: injecting into 8 layers with a control dimension of 256.

### D.3. Training Anchor-ControlNet only vs. Full-Finetuning

As ViewCrafter (Yu et al., 2024b) directly fine-tunes the entire backbone, we compare our ControlNet-based training strategy with this standard full-finetuning approach to highlight the efficiency of our design. Specifically, we encode the anchor video

*Table 6.* Different video backbones results with EPiC on RealEstate10K dataset.

| Method | Total | Subject Consist | Bg Consist | Quality Score Motion Smooth | Temporal Flicker | Aesthetic Quality | Imaging Quality | Rotation Error (↓) | Camera Score Transition Error (↓) | CamMC (↓) |
|---|---|---|---|---|---|---|---|---|---|---|
| EPiC+CogVideoX (5B) | 82.63 | 91.62 | 93.43 | 98.48 | 96.47 | 51.19 | 64.57 | 0.40 ± 0.11 | 0.86 ± 0.18 | 1.17 ± 0.23 |

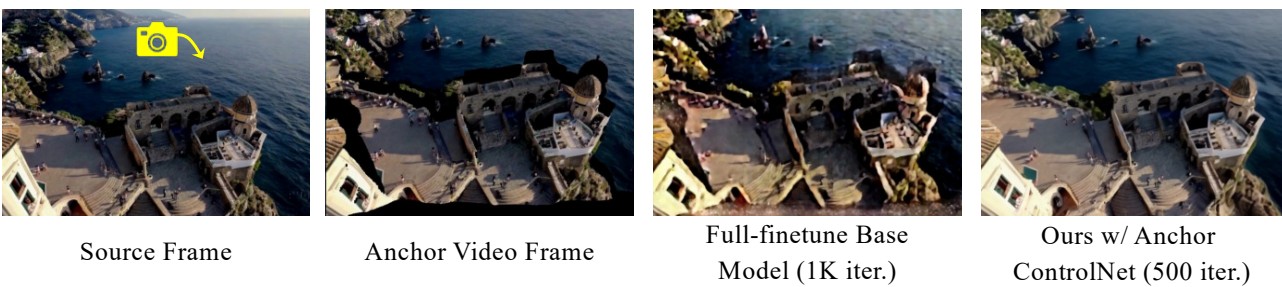

|  | 3072 | 21 | 0.42 | 0.83 | 1.19 |
| ✓ | 256 | 21 | 0.38 | 0.90 | 1.21 |
| ✗ | **256** | **8** | 0.40 | 0.86 | 1.17 |
| ✗ | 256 | 2 | 0.70 | 1.32 | 1.89 |

In contrast, our ControlNet design enables effective anchor-video conditioning without modifying the backbone, by treating the anchor video as an external control signal.

| Source Frame | Anchor Video Frame | Full-finetune Base Model (1K iter.) | Ours w/ Anchor ControlNet (500 iter.) |

*Figure 8.* Results of training with Anchor-ControlNet compared to full-finetuning.

## D.4. Additional Ablations on Anchor-ControlNet's Visibility-Aware Output Masking Design

We provide further analysis on Anchor-ControlNet's visibility-aware output masking (VAOM) design in Fig. 9. As shown, directly applying a vanilla ControlNet to the anchor video without any masking mechanism causes the model to follow errors in invisible regions, resulting in black or severely white-lined content. This indicates that a plain ControlNet architecture is insufficient for robust anchor-video conditioning. Moreover, applying VAOM only at inference time is also inadequate: it still introduces flickering in several areas, and the invisible regions fail to extend naturally from the visible scene (e.g., in the first example, the black region is completed as a brown patch). In contrast, integrating our VAOM design during both training and inference fully unlocks the base model's ability to complete invisible regions smoothly and coherently, yielding stable, clean, and artifact-free results. This unified training-time integration also enables EPiC to generalize to arbitrary masked anchor videos at test time (Fig. 2), supporting both static and dynamic settings with user-specified dynamic regions.

## D.5. Generalization to Different Backbones

We provide additional results to demonstrate EPiC's generalization across different backbones. Specifically, we select Wan-2.1-I2V-14B-480P as the backbone and train EPiC using the same settings. We evaluate the model on the RealEstate10K dataset, and report quantitative results in Tab. 6 and qualitative examples in Fig. 10. As shown, the Wan backbone yields better visual quality while maintaining comparable camera-control accuracy, demonstrating that EPiC generalizes well to stronger base models.

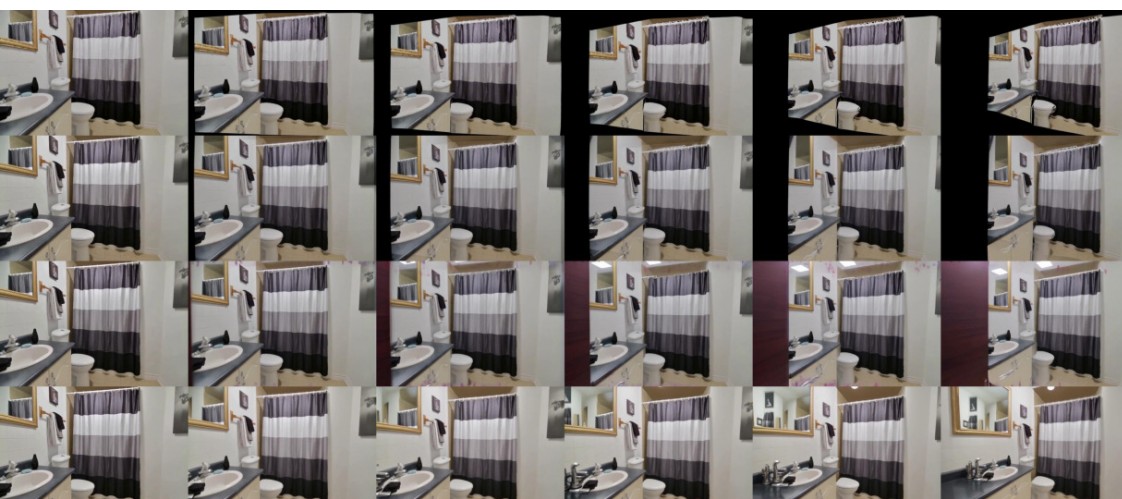

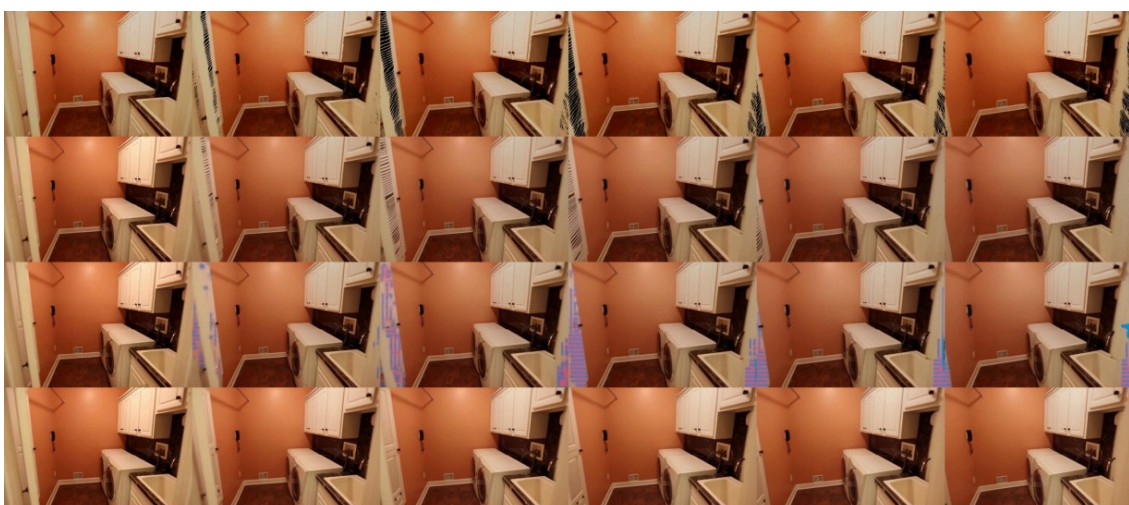

*Figure 9.* Abalations on Anchor-ControlNet's visibility-aware output masking design.

### D.6. PSNR Evaluation on RealEstate10K

In addition to camera-pose metrics and VBench scores reported in the main paper, we evaluate PSNR against ground-truth novel views on RealEstate10K. We report results on both the full test set from Table 1 and an easy subset where camera rotation $< 10$ and translation $< 0.5$ units. As shown in Tab. 8, EPiC achieves comparable or better PSNR than all baselines on both subsets.

### D.7. Robustness to Different Optical Flow Models

Our training-time anchor construction uses optical flow to estimate visibility masks. To verify that our method is not sensitive to the choice of flow model, we compare three widely used optical flow estimators: RAFT (Teed & Deng, 2020), UniMatch (Xu et al., 2023), and GMFlow (Xu et al., 2022). As shown in Tab. 9, all three flow models yield similar camera control performance, confirming that the masking-based anchor construction is robust to the choice of optical flow model.

### D.8. Scaling Up Training Data and Iterations

To evaluate the scalability of our framework, we train EPiC with a larger dataset of 30K videos from Panda70M for 5K iterations. As shown in Tab. 10, scaling up training data and iterations further improves performance on both RE10K and MiraData, demonstrating that our framework benefits from more data while already achieving strong results with minimal

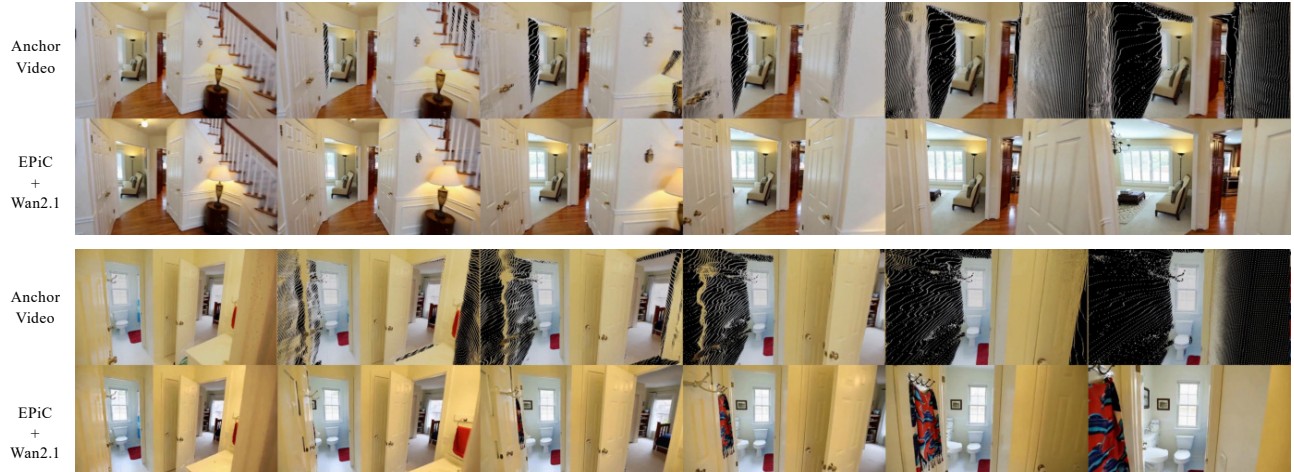

*Figure 10.* Qualitative results of EPiC with Wan2.1 Backbone on RealEstate10k.

*Table 8.* PSNR evaluation on RealEstate10K.

| Method | Full Set ↑ | Easy Set ↑ |
|---|---|---|
| CameraCtrl (He et al., 2024) | 12.06 | 15.34 |
| AC3D (Bahmani et al., 2024a) | 14.30 | 18.34 |
| FloVD (Jin et al., 2025) | 14.45 | 18.52 |
| ViewCrafter (Yu et al., 2024b) | 14.91 | 19.36 |
| Gen3C (Ren et al., 2025) | 15.42 | **19.93** |
| EPiC (Ours) | **15.51** | 19.91 |

resources.

## E. Robustness to Different Random Seeds

We demonstrate the robustness of our method in Fig. 11. Given a conditioned image, we use a specific object (highlighted with a white box) as the reference for spatial consistency. For AC3D, varying the random seed leads to noticeable changes in the spatial positions of other objects (highlighted in red boxes). This is especially evident in Seed 3, where the generated object's position drifts significantly from the reference, failing to maintain spatial alignment. In contrast, our method consistently preserves the spatial relationship across different seeds. The objects in our generated videos (highlighted in green boxes) remain stable and aligned with the referenced object, demonstrating strong robustness to seed variation.

## F. Qualitative Comparison with Baselines

### F.1. Image-to-Video Camera Control

**With ViewCrafter.** We provide qualitative comparisons in Fig. 12. While both methods can follow the anchor video, ViewCrafter's visual quality is noticeably lower: in RealEstate10K, it gradually turns a table into a sofa in the first example and makes the toy bear disappear in the second; on MiraData, it often generates messy and unrealistic humans. More examples can be found on our website.

**With FloVD.** We provide qualitative comparisons in Fig. 13. Both EPiC and FloVD share the same CogVideoX-5B-I2V backbone, and their visual quality is generally comparable. However, FloVD struggles to follow the camera trajectory as accurately as ours. We attribute this to its indirect flow-map–based conditioning and the flow-based condition-output misalignment introduced during training. More examples can be found on our website.

*Table 9.* Robustness to different optical flow models on RE10K.

| Flow Model | RotErr ↓ | TransErr ↓ | CamMC ↓ |
|---|---|---|---|
| RAFT (Teed & Deng, 2020) | **0.40** ± 0.11 | 0.86 ± 0.18 | **1.17** ± 0.23 |
| UniMatch (Xu et al., 2023) | 0.42 ± 0.12 | **0.85** ± 0.19 | 1.19 ± 0.25 |
| GMFlow (Xu et al., 2022) | **0.40** ± 0.13 | 0.88 ± 0.21 | 1.19 ± 0.27 |

*Table 10.* Effect of scaling up training data and iterations.

| Dataset | #Vids | VQ ↑ | RotErr ↓ | TransErr ↓ | CamMC ↓ |
|---|---|---|---|---|---|
| RE10K | 5K | 82.63 | 0.40 | 0.86 | 1.17 |
| RE10K | 30K | **82.70** | **0.34** | **0.83** | **1.01** |
| MIRA | 5K | 82.89 | 0.66 | 1.78 | 2.10 |
| MIRA | 30K | **82.91** | **0.60** | **1.69** | **2.01** |

**With Gen3C.** We provide qualitative comparisons in Fig. 14. While both methods can follow the anchor video, Gen3C's visual quality is noticeably lower on MiraData. We attribute this to its training data: Gen3C is trained heavily on scene-level datasets, which makes the model behave like a scene-level NVS system and generalize poorly to more dynamic, human-centric content. More examples can be found on our website.

**Controllable Dynamic Objects.** As shown in the examples in Fig. 21, EPiC flexibly supports both dynamic and static scenes in I2V. By contrast, FloVD mainly handles dynamic objects, and Gen3C supports only static scenes. EPiC can naturally do both by simply adjusting the mask in the anchor video to specify which regions should move and which should stay fixed.

### F.2. Video-to-Video Camera Control

**With Gen3C and TrajectoryCrafter.** We provide qualitative comparisons in Fig. 15. In the first example, both Gen3C and TrajectoryCrafter follow the anchor video too rigidly, resulting in a half-body mammoth or incorrect occlusions caused by erroneous anchor-video rendering. We attribute this to their full-finetuning strategy, which turns the models into strict anchor-following systems with weakened semantic priors. In contrast, EPiC follows the anchor video while still generating semantically coherent content, thanks to its frozen-backbone design that preserves strong first-frame semantic priors. More examples can be found on our website.

**With ReCamMaster.** We provide qualitative comparisons in Fig. 15. We observe several issues with RecamMaster (1) Without explicit 3D guidance, it struggles to maintain correct geometry, as shown in the first example where the selfie stick becomes distorted; (2)As its conditioning is based on absolute camera parameters, it fails on videos with camera motion (second example), causing both the moving camera and the SUV to appear static; (3) it hallucinates objects not present in the source video (third example), such as an extra basketball and even a nonexistent backboard; and (4) it sometimes produces oil-painting-like artifacts (fourth example). In contrast, EPiC generates more natural and stable results without these issues, thanks to the explicit anchor-video guidance and the strongly maintained first-frame semantic prior. More examples can be found on our website.

## G. Additional Qualitative Results

**I2V Qualitative Examples.** We showcase diverse qualitative examples of I2V camera control spanning a wide variety of scenarios in Fig. 17, including daily-life activities (cooking, dining, exercising), human–animal interactions (fox resting, horse walking), transportation (cycling, subway), outdoor navigation (kayaking, hiking, urban scenes), and complex virtual environments (video games, historical architectures, and futuristic cityscapes). These examples highlight that EPiC can handle both indoor and outdoor scenes, real-world and synthetic data, and static as well as dynamic objects. The results demonstrate strong generalization across highly diverse contexts, producing coherent motion and faithful camera control without overfitting to specific domains. More examples can be found on our website.

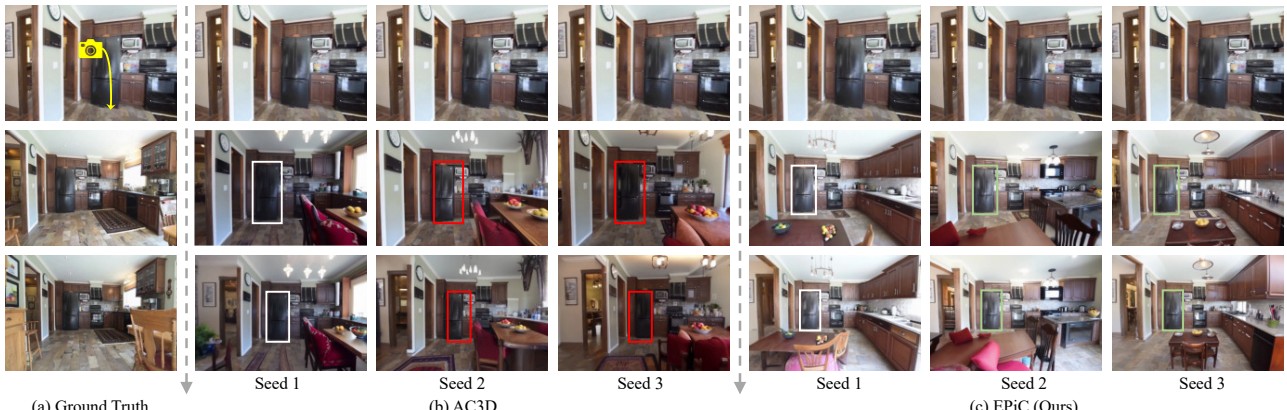

*Figure 11.* Robustness to different random seeds

**V2V Qualitative Examples.** We present diverse examples of V2V camera control spanning movie clips and in-the-wild videos in Fig. 18 and Fig. 19. Across various camera trajectories, our method is able to faithfully follow the target motion while producing high-quality and visually coherent results. More examples can be found on our website.

**V2V Multi-Camera Shooting.** We further demonstrate multi-camera shooting in Fig. 20, where multiple trajectories are generated from a single input video. The results show strong temporal consistency across different camera views, indicating that our method can maintain coherent scene structure and appearance under diverse camera motions. More examples can be found on our website.

**I2V Inference Modes.** We show results of different I2V inference modes (mode (b) and (c) in Fig. 2) in Fig. 21. With the full point cloud in mode (b), our method tends to generate static content. By masking the point cloud in mode (c), we can make specific objects or background dynamic, demonstrating the ability to control both object motion and scene dynamics. More examples can be found on our website.

**Examples of Constructed Anchor Videos.** We present examples of high-quality anchor videos constructed from Panda70M source videos in Fig. 22. Our method consistently maintains spatial coherence and masks regions that were initially not visible in the first frame, even when objects exhibit significant movements across frames, while the Panda70M provides both diverse and dynamic video data. Such high-quality and diverse anchor videos further help the efficient learning by our model. Video examples can be found on our website.

## H. Additional Applications: Fine-Grained Control

We present several additional applications demonstrating different types of fine-grained control based on a single image with our anchor-video conditioning.

**Text-Guided Scene Control.** Our model effectively demonstrates dynamic text-guided video generation capabilities, enabling flexible scene synthesis across different styles while maintaining temporal and spatial consistency. Fig. 23 illustrates examples of our text-guided scene control. Starting from an initial frame with a fixed forward camera trajectory, our method generates subsequent video frames conditioned on different textual prompts. The newly prompted objects are introduced into the generated scene (highlighted in red text and boxes), while the objects present in the initial frame remain consistently visible throughout the video (highlighted in green text and boxes).

**Object 3D Trajectory Control via Anchor Video Manipulation.** We also demonstrate the flexibility of our method in enabling 3D trajectory control for objects. The input is usually a 3D trajectory (*e.g.*, indicating moving backwards with 2 meters) applied to a specific object (*e.g.* corgi). We encode the desired motion into the anchor video by manipulating it based on the 3D trajectory. Specifically, following a similar approach to our inference setup with masked point clouds, we use GroundedSAM (Ren et al., 2024) to obtain the segmentation mask of the corgi, extract the point cloud corresponding to

the corgi, and isolate the background point cloud without the corgi. We then simulate motion by translating the corgi's point cloud backward by 2 meters relative to the background over time (we don't move the background point cloud), producing a dynamic point cloud sequence for rendering. In this setup, we focus solely on trajectory control, thus, we remain the camera trajectory static during rendering. The resulting anchor video depicts the corgi moving backward and serves as strong guidance. Our results are illustrated in Fig. 24, where our approach successfully generates scenarios in which the corgi steps backward. In contrast, AC3D, which conditions only on camera embeddings, which lack explicit trajectory information, fails to generate this backward motion even with "stepping backward" included in the textual condition. This comparison highlights the strength of our method in interpreting and executing precise object-level movements in 3D space, showcasing its superior capability for controllable video generation.

**Regional Animation.** Our method is also applicable to regional image animation, where motion is localized to a specific area based on a short text prompt and a user-provided click or prior mask. To achieve this, we directly create the anchor video by repeating the source image and applying the regional mask to each frame. As shown in Fig.25 (a), given the prompt "the corgi shakes its head," with corresponding corgi head mask, our method generates a video in which only the corgi's head moves while the rest of its body remains still, accurately following both the textual instruction and the specified region. In contrast, Fig.25 (b) highlights a failure case of AC3D—when the intended motion is for the palm tree to move, AC3D incorrectly animates the corgi instead. Our method, however, successfully isolates and animates the palm tree, demonstrating its ability to localize motion precisely based on regional guidance and text. This showcases the fine-grained spatial control ability enabled by our approach.

## I. Failure Analysis

Since our model learns to follow the anchor video in visible regions, it can be affected when the estimated point-cloud structure or occlusion masks are inaccurate. We provide two examples in Fig. 26 on the website illustrating the main failure modes: (1) **Incorrect point-cloud structure.** In the first example, a misestimated point cloud causes the man with a backpack in the anchor video to appear tilted, and our result partially inherits this (e.g., a slightly stretched neck). The face of the person next to him also begins to tilt. In comparison, ViewCrafter loses track of the motion and produces randomly distorted humans, while Gen3C strictly follows the erroneous structure, resulting in even more distorted outputs. EPiC, despite inheriting some of the structural bias, remains noticeably more stable. (2) **Incorrect occlusion.** In the second example, background color leaks through the kangaroo's face in the anchor video. EPiC interprets this as a mild blue lighting effect, whereas TrajectoryCrafter and Gen3C rigidly copy the artifact and produce visible holes in the face. These analyses clarify how EPiC behaves under imperfect 3D estimation and demonstrate that—even in failure cases—it remains more robust than baseline methods.

## J. Limitations and Broader Impacts

Our training-time anchor construction relies on optical flow, which may produce inaccurate visibility masks under fast motion or heavy occlusion. Additionally, while training is 3D-free, test-time inference still depends on depth estimation for point-cloud-based anchor rendering, inheriting its limitations for challenging viewpoint changes.

EPiC trains a lightweight adapter on a backbone video diffusion model. As such, its performance, output quality, and potential visual artifacts are inherently influenced by the capabilities and limitations of the underlying backbone models it relies on. For instance, if the backbone model struggles with generating complex, rare, or previously unseen scenes and objects, then EPiC may also exhibit suboptimal generation results. This dependency highlights the importance of selecting strong and reliable backbone models when applying EPiC.

While EPiC can benefit numerous applications in video generation, similar to other visual generation frameworks, it can also be used for potentially harmful purposes (e.g., creating false information or misleading videos). Therefore, it should be used with caution in real-world applications.

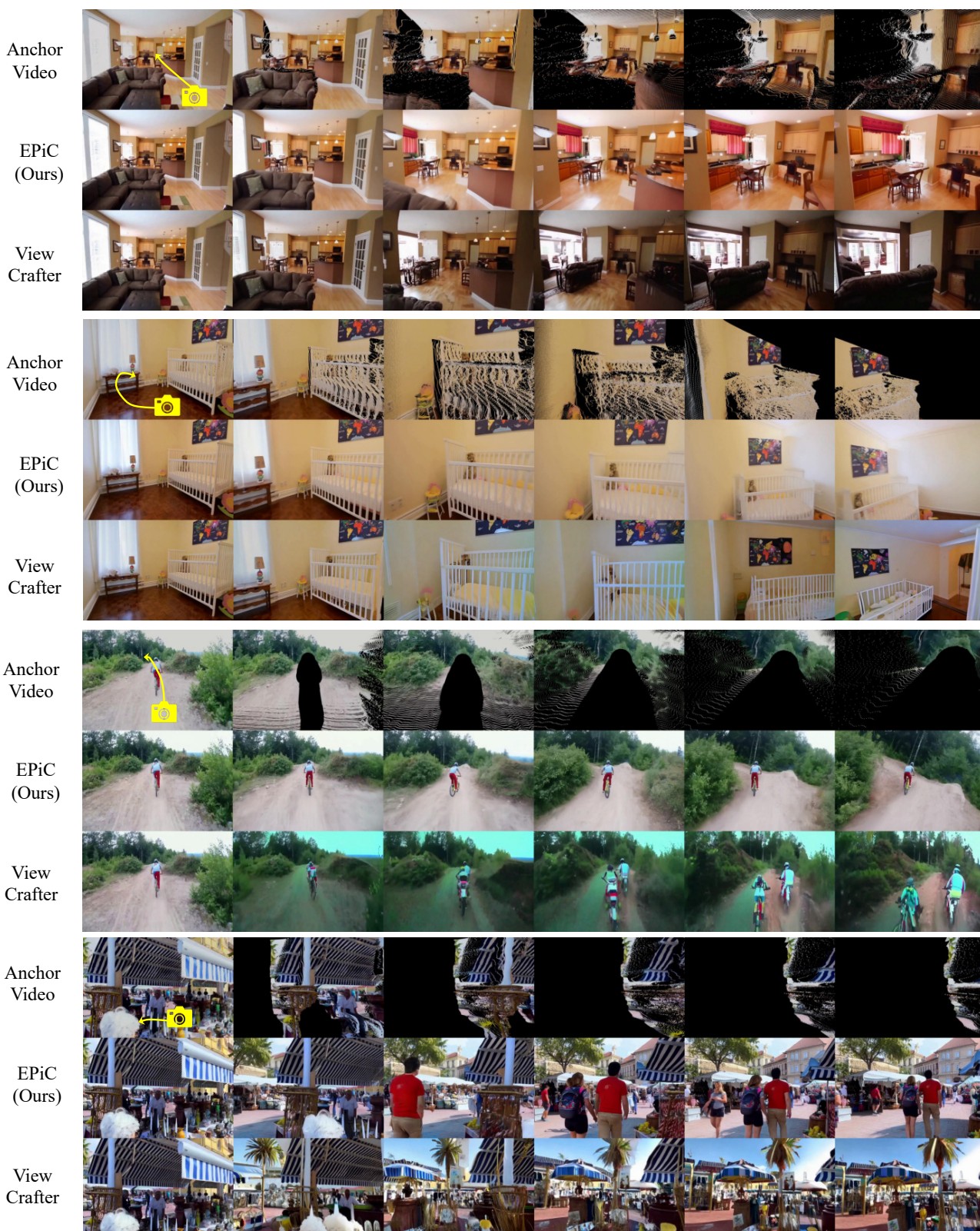

*Figure 12.* I2V Comparison with ViewCrafter. The first two examples are from RealEstate10K, while the last two examples come from MiRaData.

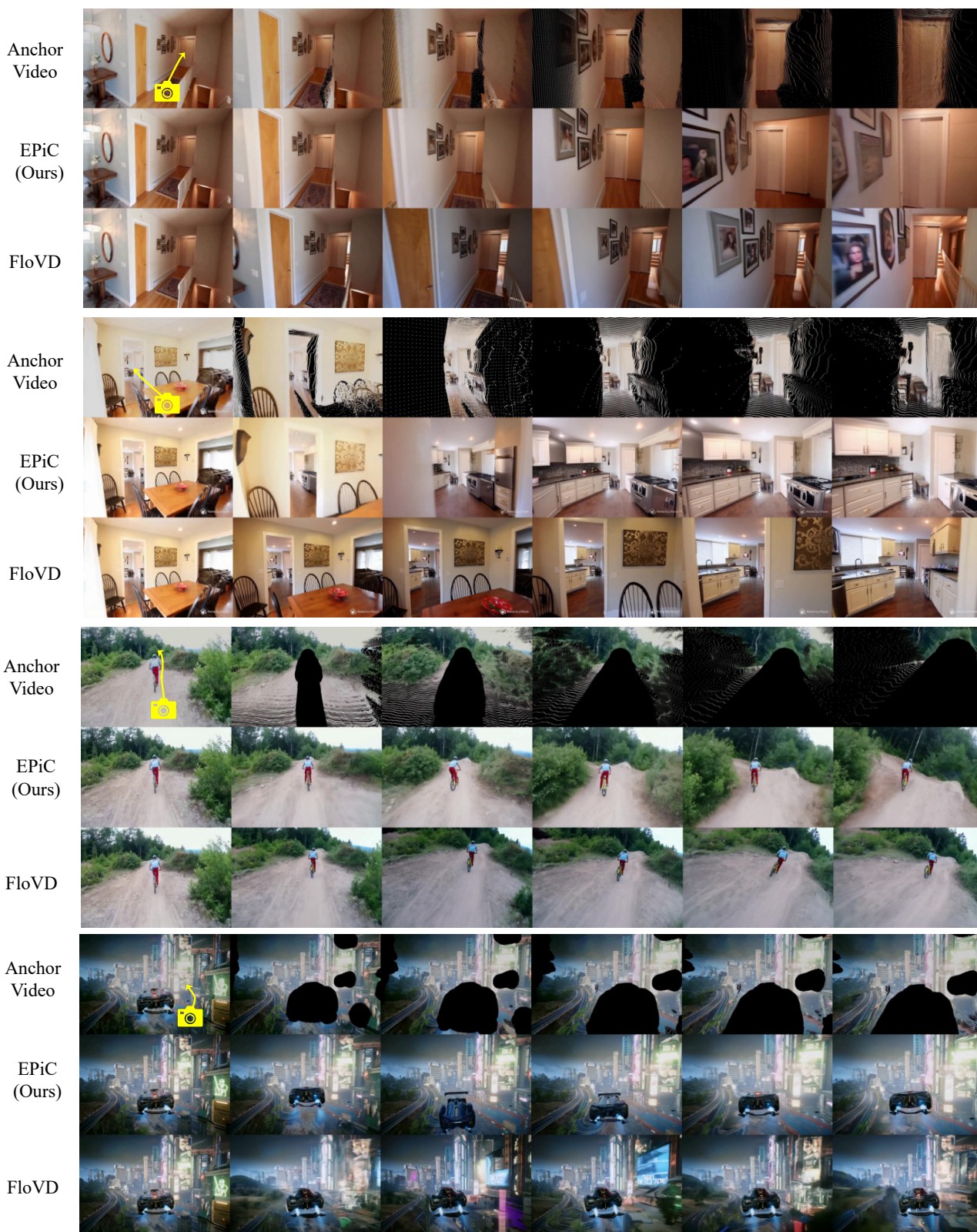

*Figure 13*. I2V Comparison with FloVD. The first two examples are from RealEstate10K, while the last two examples come from MiRaData.

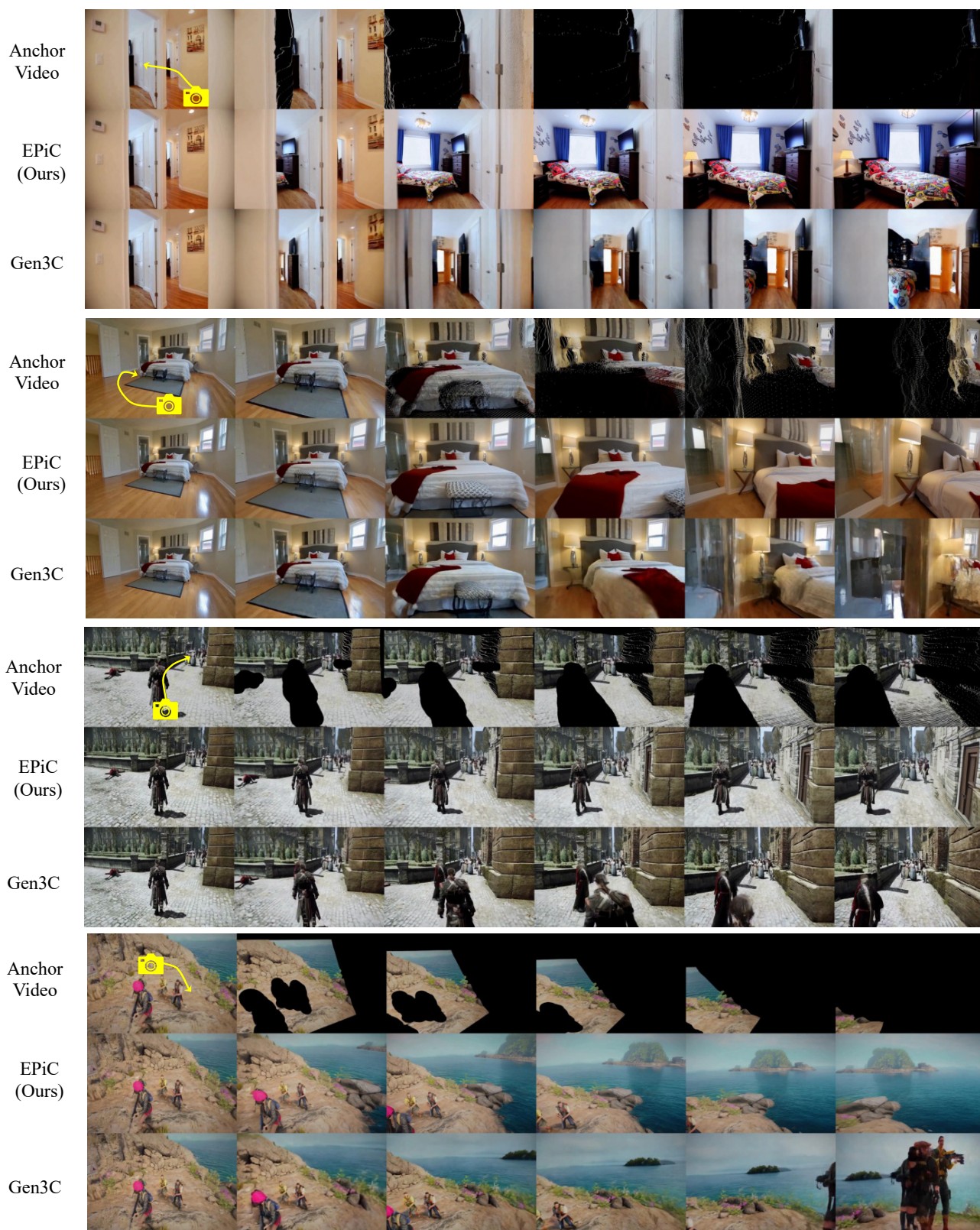

*Figure 14.* I2V Comparison with Gen3C. The first two examples are from RealEstate10K, while the last two examples come from MiRaData.

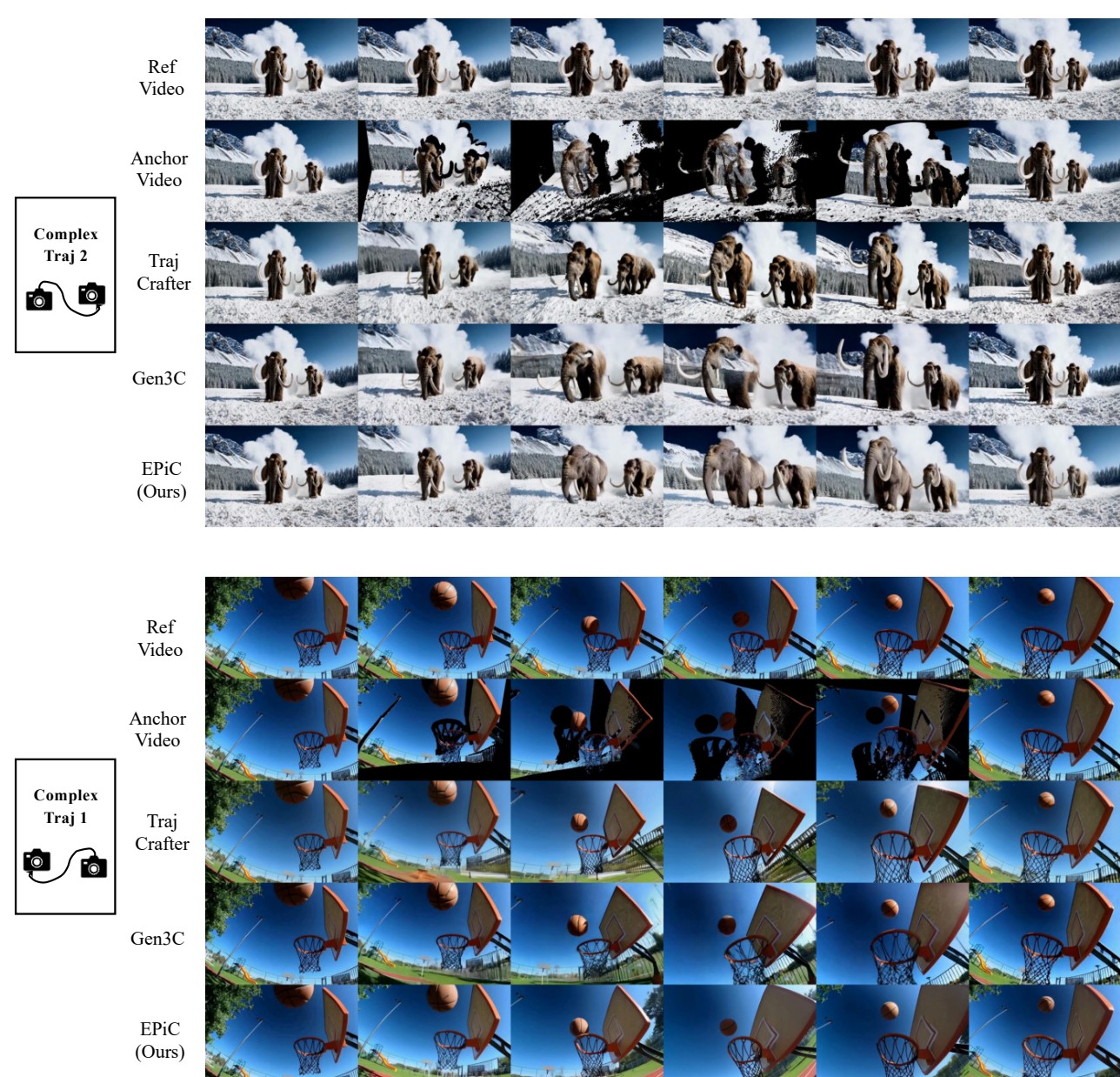

*Figure 15.* V2V Comparison with Gen3C and TrajectoryCrafter.

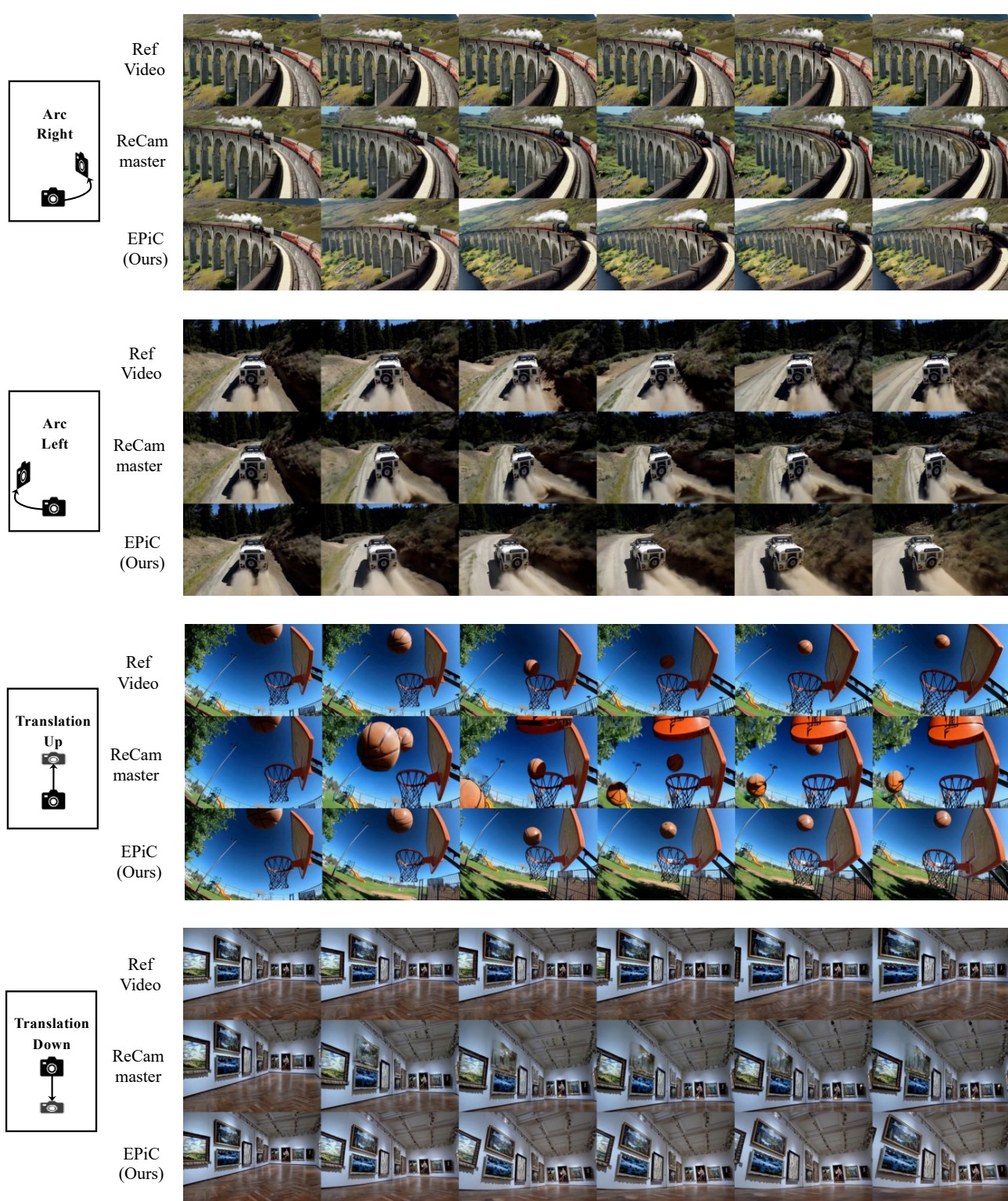

*Figure 16.* V2V Comparison with ReCamMaster.

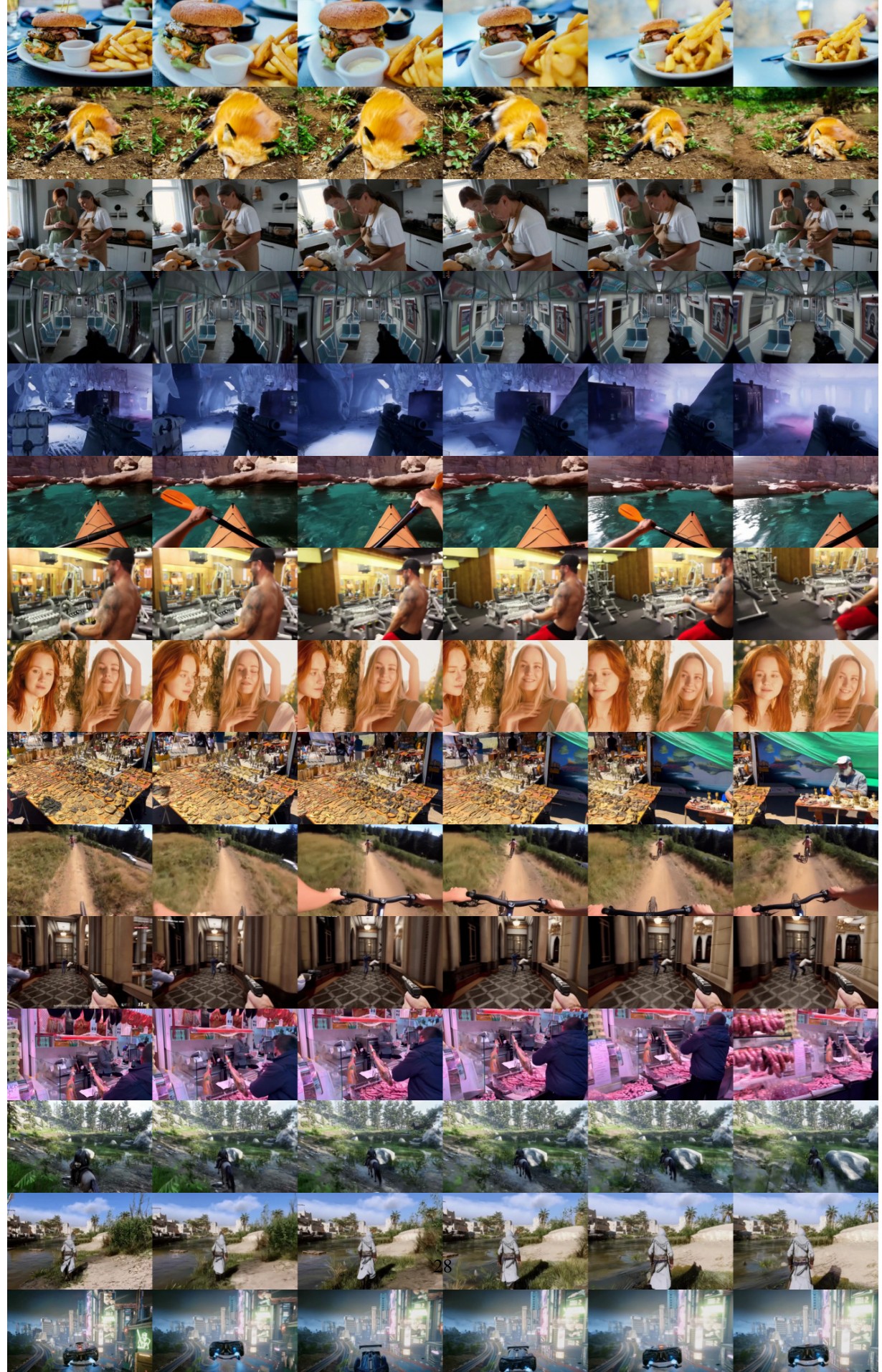

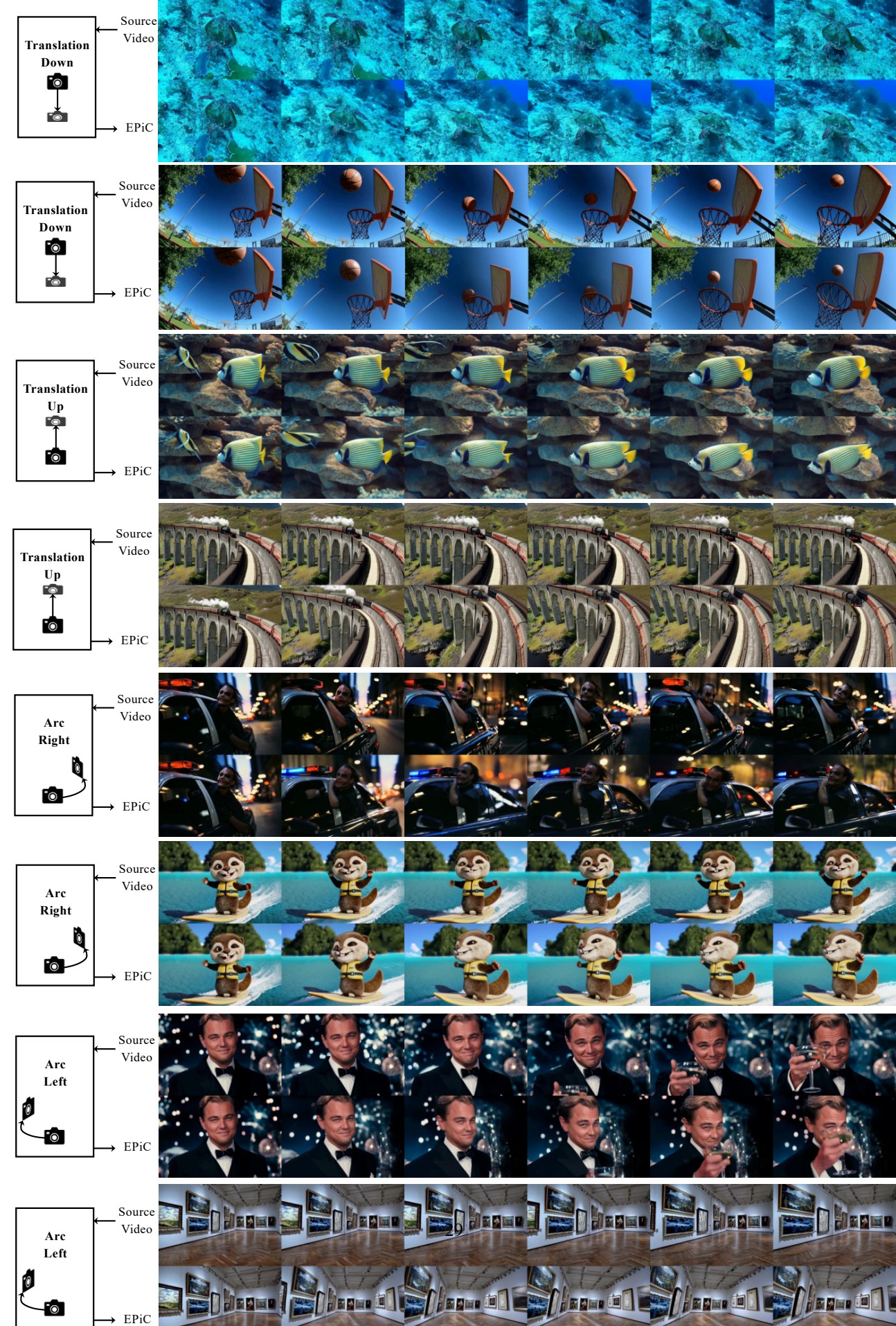

**EPiC: Efficient Video Camera Control Learning with Precise Anchor-Video Guidance**

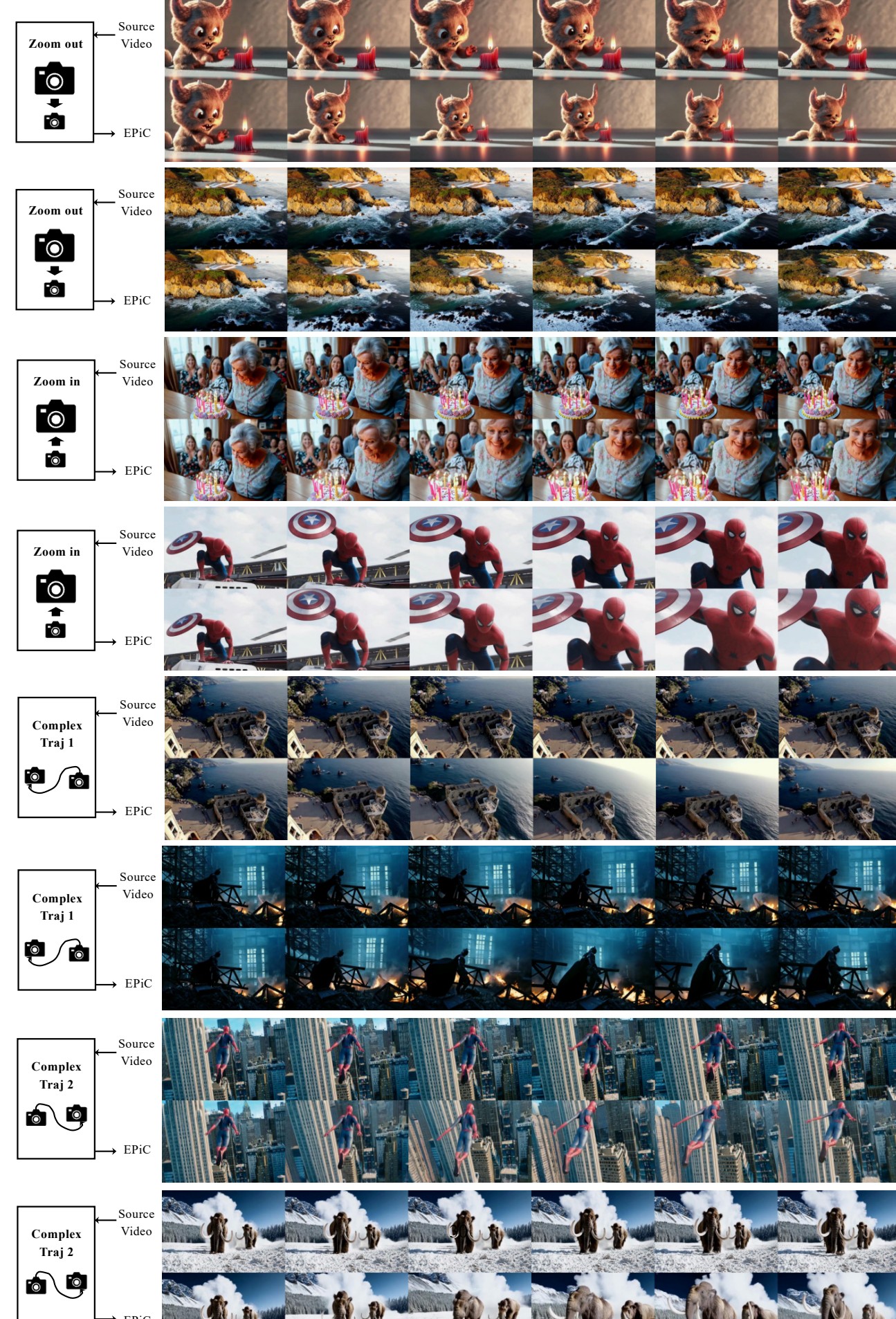

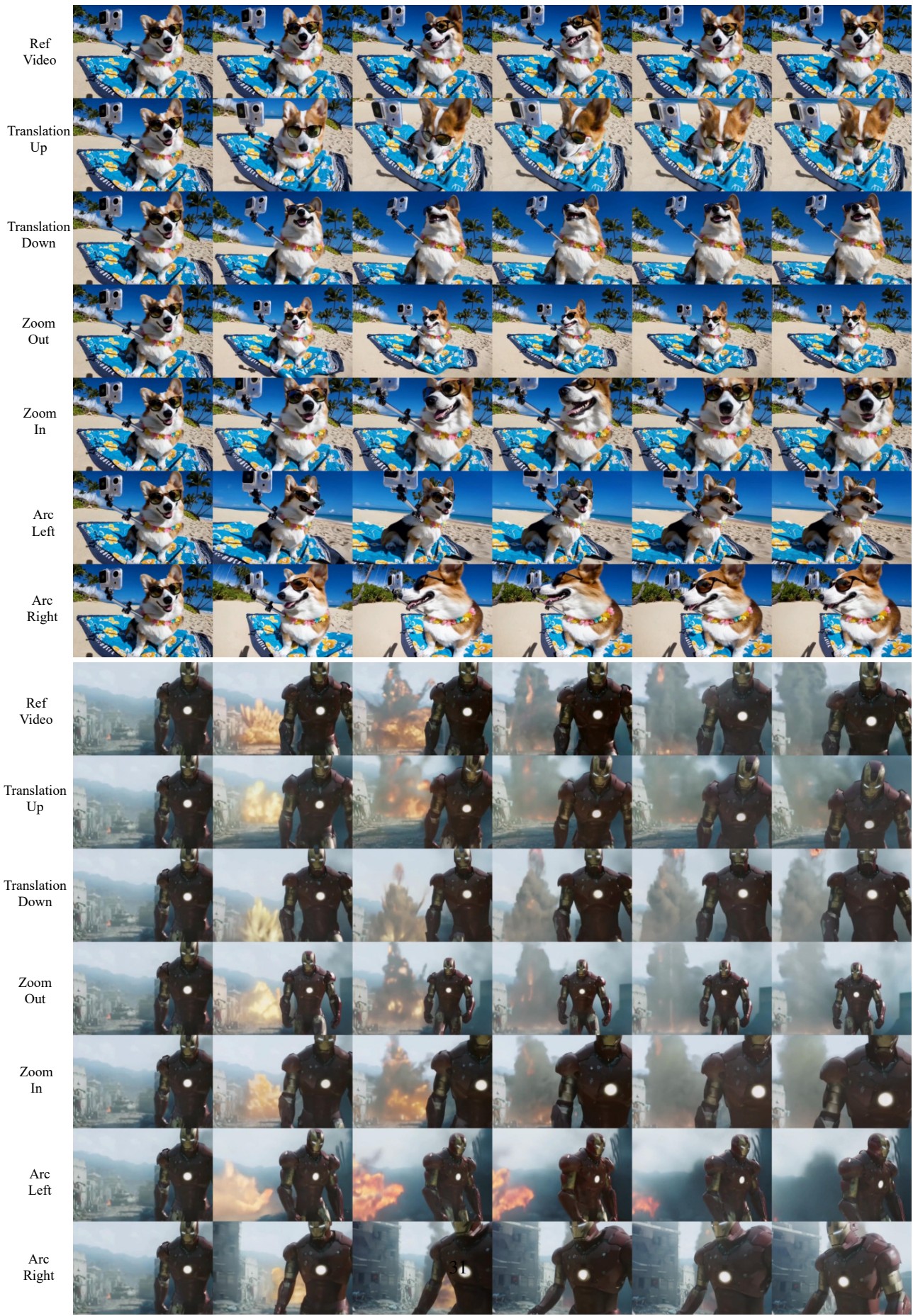

*Figure 20.* Multi-camera shooting examples for V2V.

EPiC Mode (b)
Anchor Video
(Full Static)

EPiC Mode (b)
Generated Video
(Full Static)

EPiC Mode (c)
Anchor Video
(Dynamic
Foreground)

EPiC Mode (c)
Generated Video
(Dynamic
Foreground)

EPiC Mode (c)
Anchor Video
(Dynamic
Background)

EPiC Mode (c)
Generated Video
(Dynamic
Background)

FloVD
Generated Video
(Dynamic)

Gen3C
Generated Video
(Static)

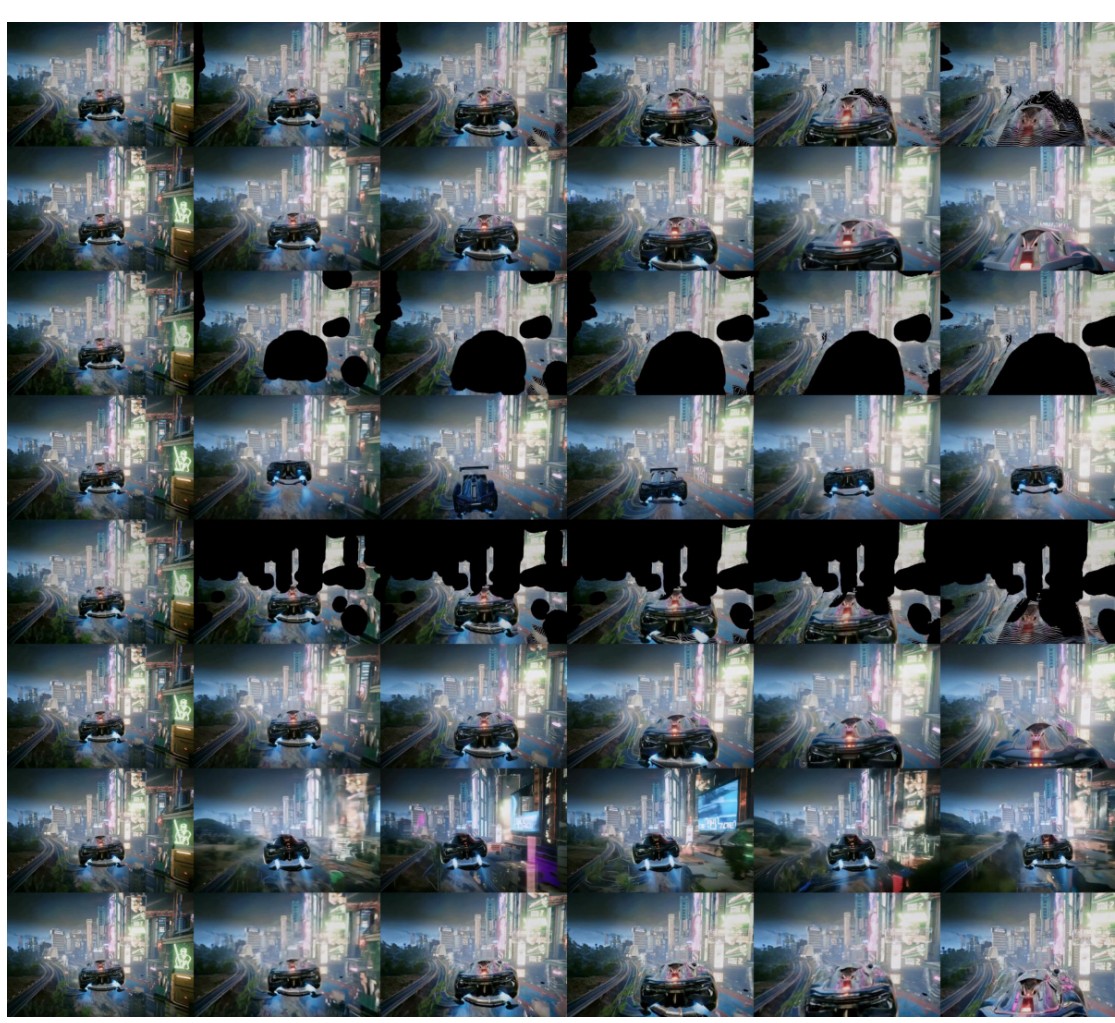

EPiC Mode (b)
Anchor Video
(Full Static)

EPiC Mode (b)
Generated Video
(Full Static)

EPiC Mode (c)
Anchor Video
(Dynamic
Foreground)

EPiC Mode (c)
Generated Video
(Dynamic
Foreground)

EPiC Mode (c)
Anchor Video
(Dynamic
Background)

EPiC Mode (c)
Generated Video
(Dynamic
Background)

FloVD
Generated Video
(Dynamic)

Gen3C

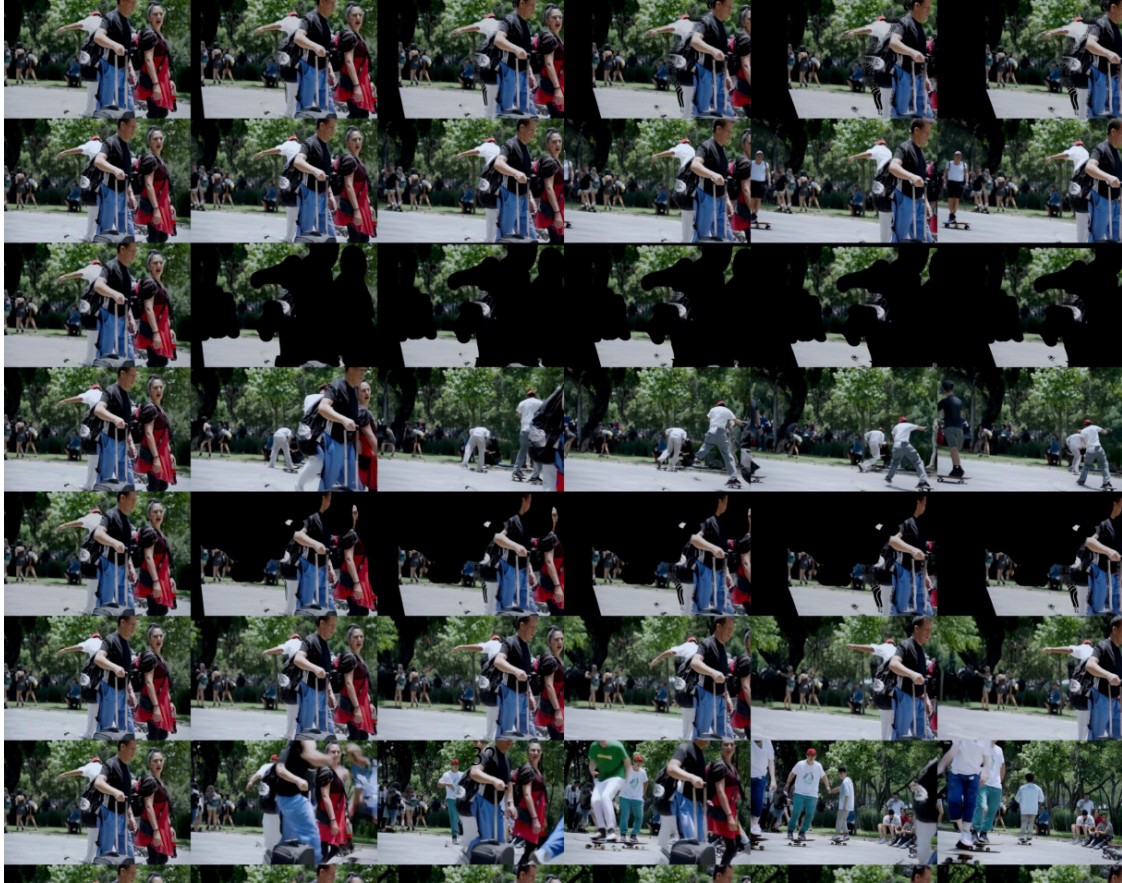

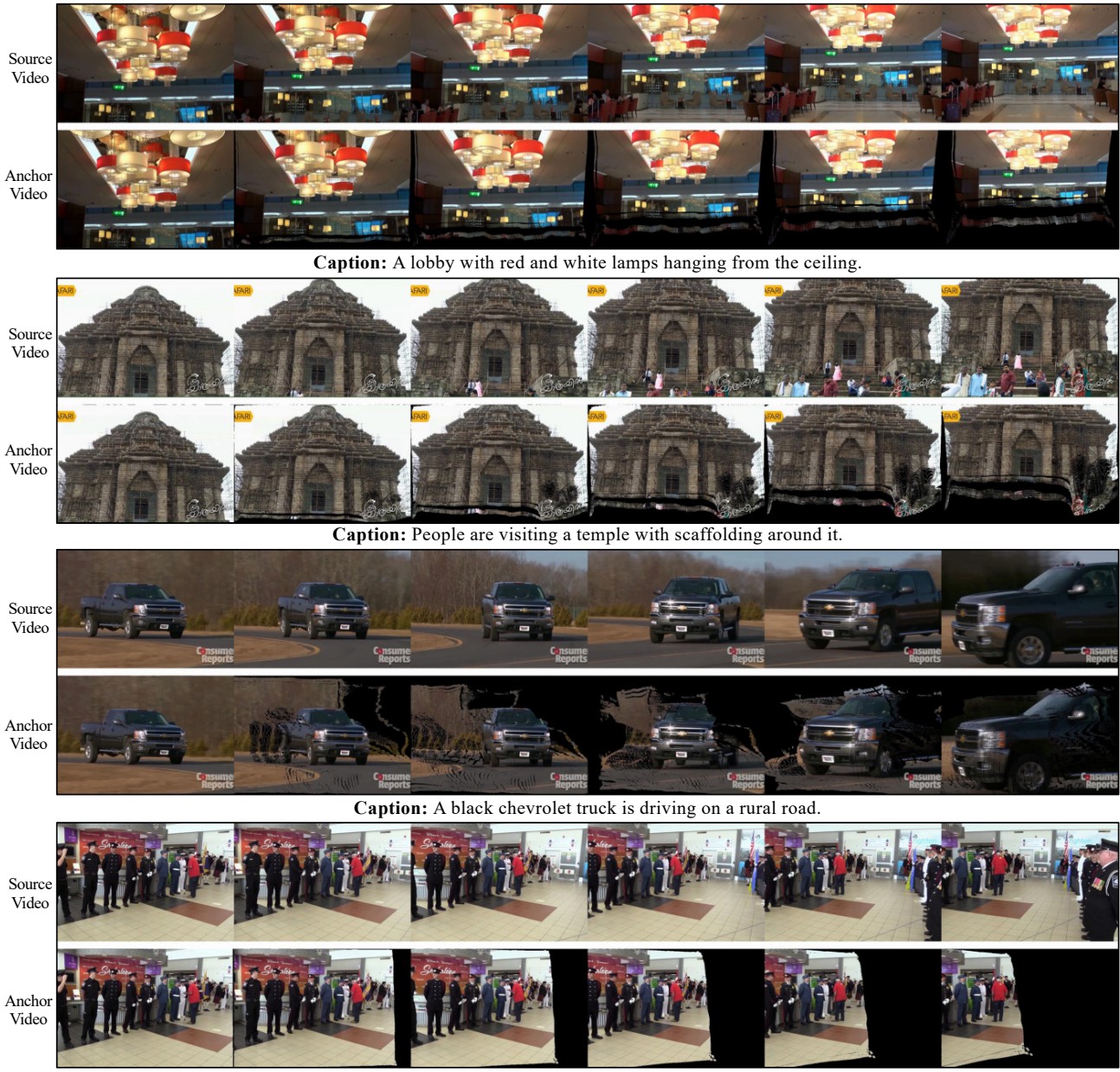

*Figure 22.* Examples of constructed anchor videos. The source video and corresponding captions are obtained from Panda70M.

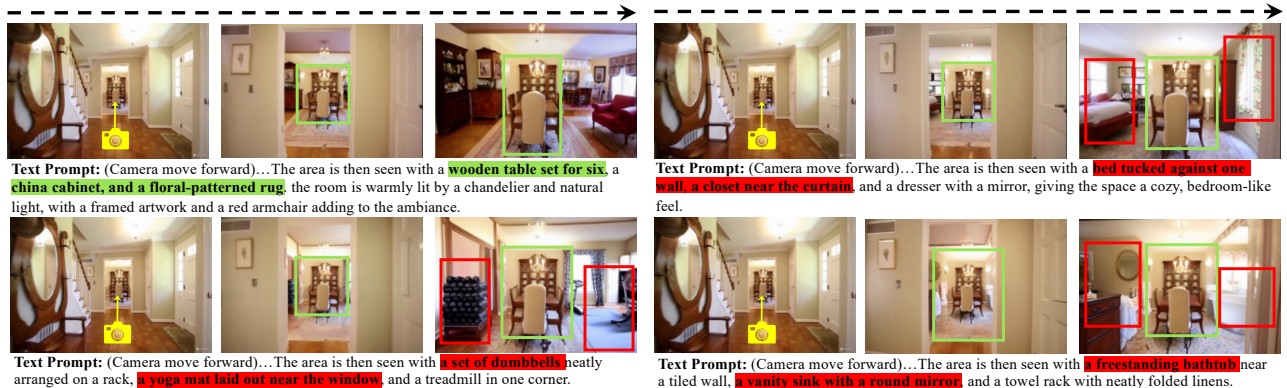

Figure 23. Examples of text-guided scene control.

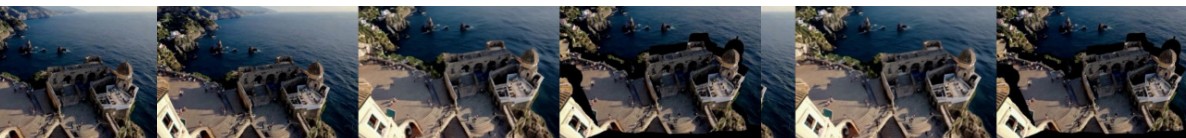

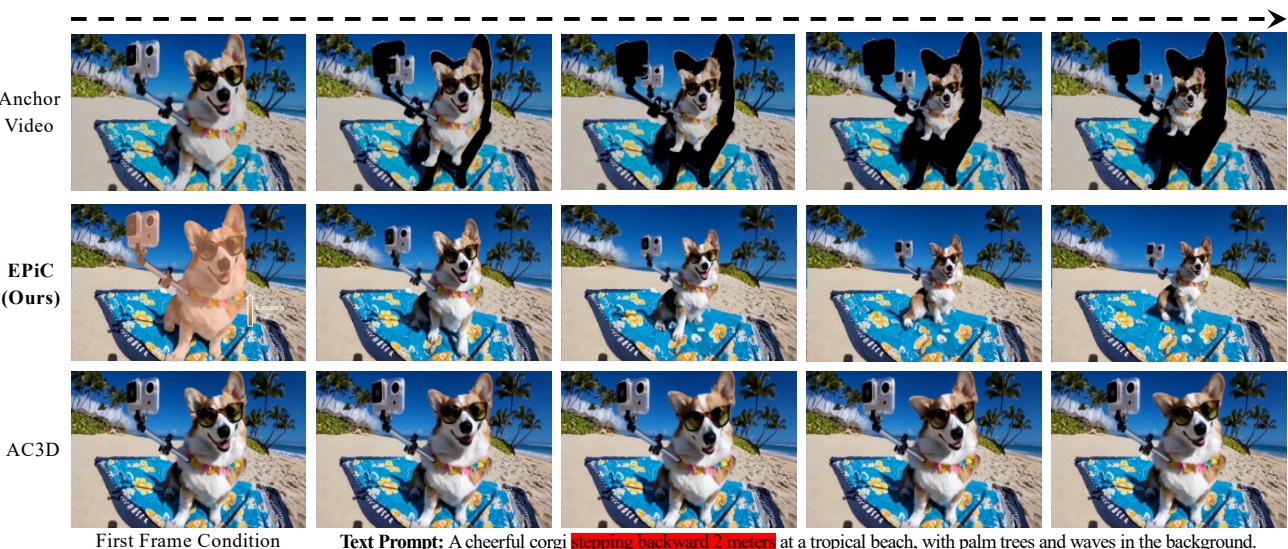

Figure 24. Examples of object 3D trajectory control via anchor video manipulation.

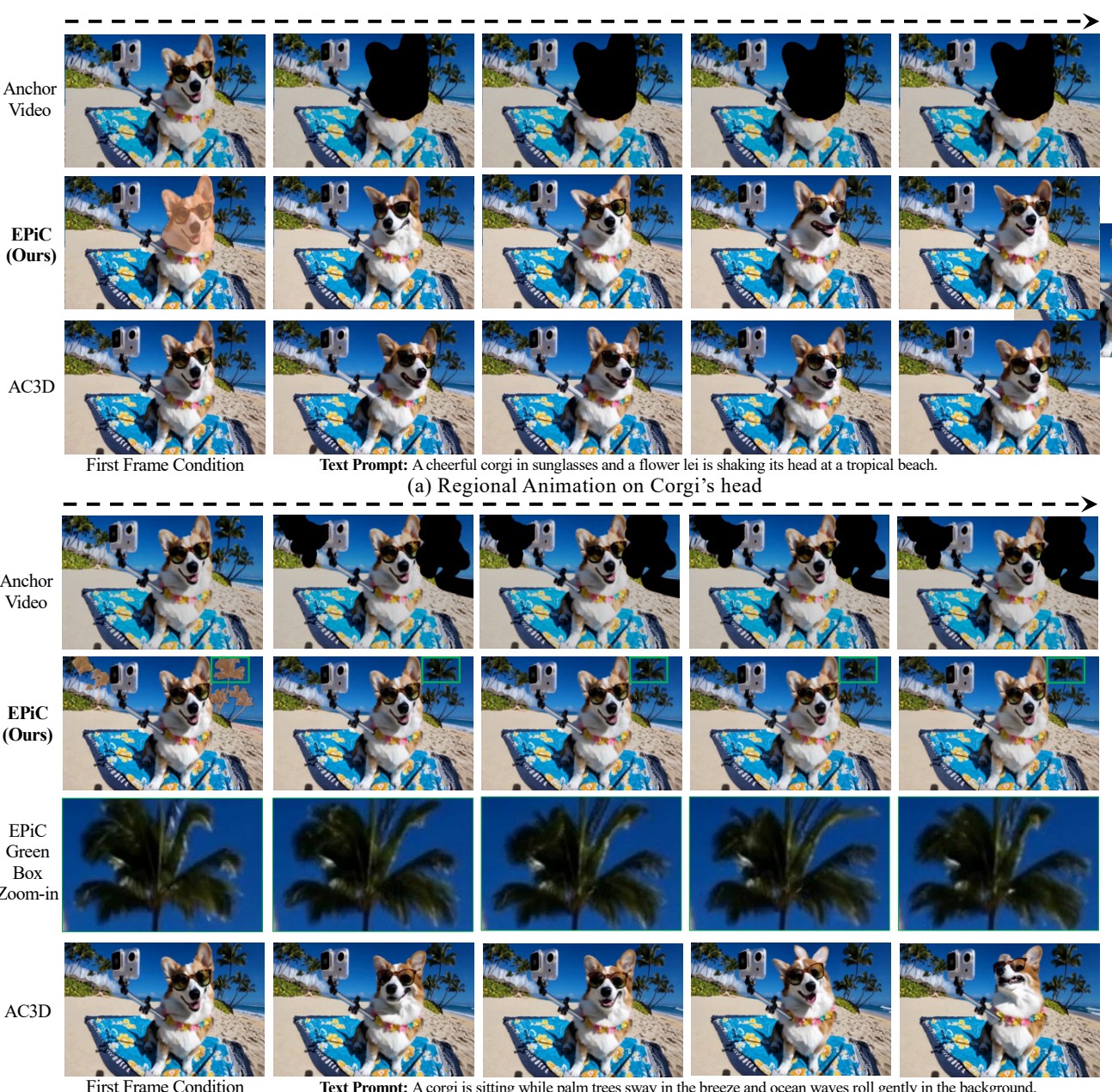

*Figure 25.* Examples of Regional Animation.

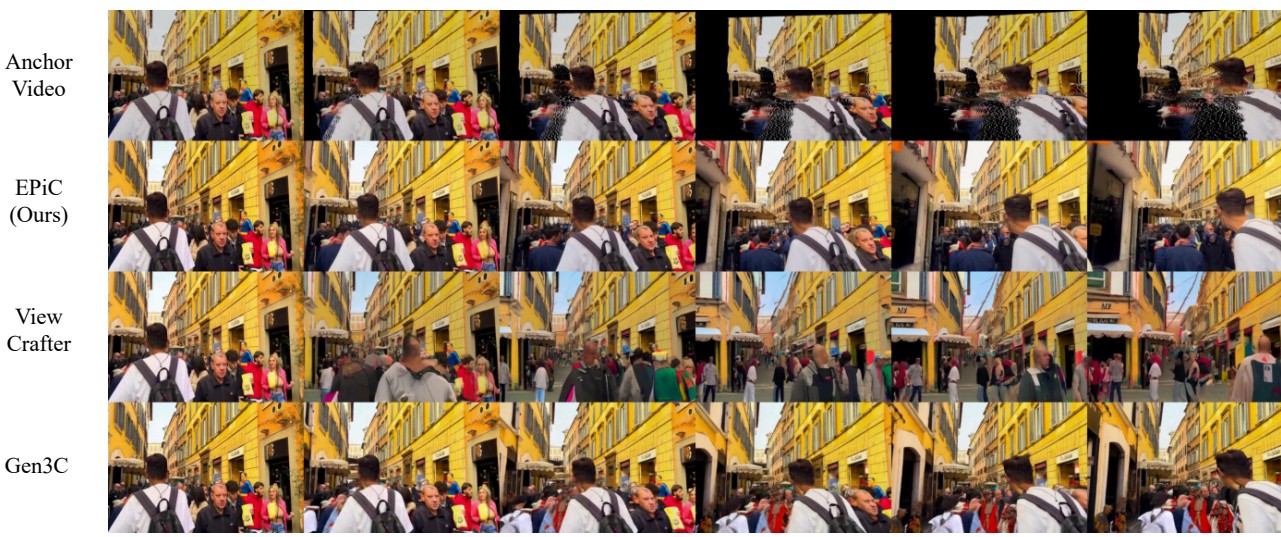

**Failure case 1: Incorrect Point Cloud Structure**

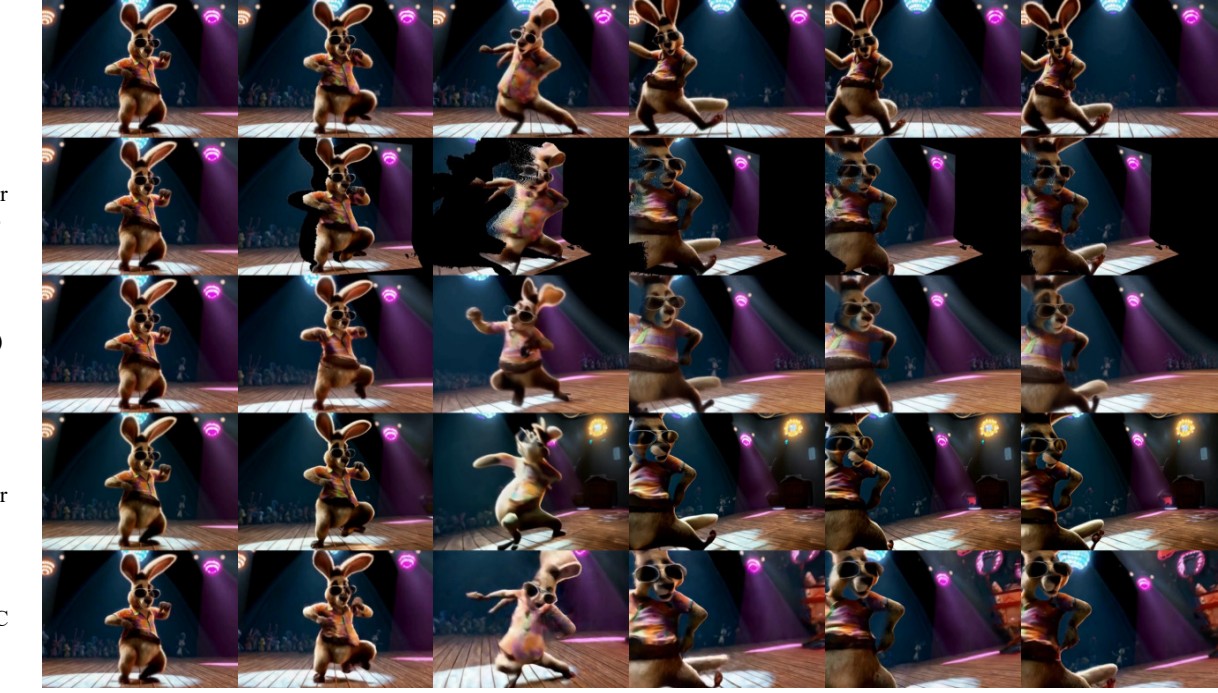

**Failure case 2: Incorrect Occlusion**

*Figure 26.* EPiC failure cases with baseline comparison.

