# OpenReview forum: "EPiC: Efficient Video Camera Control Learning with Precise Anchor-Video Guidance"
_ICML.cc/2026/Conference — ICML 2026 regular_

### Official Review · Reviewer_hbsf · 2026-03-01

**Soundness:** 3
**Presentation:** 3
**Significance:** 3
**Originality:** 3
**Overall Recommendation:** 5
**Confidence:** 4

**Summary:**

EPiC addresses camera-controlled video generation via anchor-video guidance, without relying on explicit camera pose or point-cloud estimation. It constructs precisely aligned training anchors by visibility-masking the source video using first-frame correspondences (dense optical flow), ensuring consistency in visible regions while masking newly revealed areas. A lightweight Anchor-ControlNet (<1% extra parameters) operates only on visible regions, leaving disocclusions to the frozen base diffusion model. Trained with minimal compute and data, EPiC achieves strong camera-control accuracy on RealEstate10K and MiraData, and supports zero-shot transfer to V2V settings.

**Compliance With Llm Reviewing Policy:**

Affirmed.

**Final Justification:**

The rebuttal addressed my main concerns, I raise my score to accept.

**Key Questions For Authors:**

1. How does performance change if the optical flow is degraded: that is, different flow models, synthetic noise, and low-texture scenes? A small robustness plot would be very helpful.
2. For V2V, can you add a more systematic benchmark, or at least prompt/scene categories + success rate, and a short failure-mode analysis?
3. What is the wall-clock overhead of anchor construction, which is flow/masking/artifact injection, relative to training/inference?
4. How sensitive is Anchor-ControlNet placement (first 25% layers) and the visibility-mask fusion to step count / sampler?

**Limitations:**

The impact statement is reasonable, but the paper should more explicitly discuss limitations around flow failures, large viewpoint changes, and cases where “visible-only control” may be insufficient, e.g., when camera motion heavily depends on newly revealed geometry.

**Strengths And Weaknesses:**

## Strengths
* The visibility-masked anchors directly target the alignment problem that makes anchor-guided training expensive and unstable.
* Masking ControlNet’s effect on visible regions is a simple, convincing separation of responsibilities.
* Efficiency claims are unusually strong and backed by numbers/ablations (tiny training set, few steps, small adapter).
* Broad evaluation for the task, i.e. I2V + some V2V, multiple baselines, camera metrics + VBench.

## Weaknesses
* The “freeze mask when too small” heuristic is sensible but suggests failure modes under fast motion/occlusion; I’d like more quantified robustness vs flow errors and trajectory magnitude.
* Given the strong “zero-shot V2V” claim, more systematic evaluation and failure cases would strengthen the story, perhaps some qualitative + Kubric4D metrics
* Baseline fairness details, that is, the same base model, test-time anchor type, and compute budget, could be clearer; a standardized tuning protocol would help readers trust small metric gaps.

---

> ### Author Rebuttal · Authors · 2026-03-31
>
> We thank the reviewer for recognizing that our visibility-masked anchors directly target the alignment problem, the convincing separation of responsibilities in our ControlNet design, and the strong efficiency with ablations.
>
> >**W1 & Q1: Robustness to optical flow errors/mask design/trajectory magnitude.**
>
> We thank the reviewer for these important questions. We provide additional analysis below.
>
> **Optical flow robustness.**  We tested three different optical flow models:
>
> | Flow Model | CamMC (↓) |
> |---|---|
> | RAFT| 1.17±0.23 |
> | UniMatch| 1.21±0.25 |
> | GMFlow | 1.19±0.27 |
>
> Results are consistent, showing our method is robust to flow errors, as alignment is always enforced in visible regions via masking, leading to stable learning.
>
> **Freeze-mask.** This is a training-time strategy that prevents anchor guidance from becoming too sparse on extremely large motion samples. At test time, we use 3D point-cloud rendering to construct anchors, so this heuristic does not apply during inference.
>
> **Robustness to trajectory difficulty.** We split our RE10K test set into Easy (rotation<10°, translation<0.5), Medium (10°–40°, 0.5–2.0), and Hard (> 40° or > 2.0). As expected, accuracy decreases with increasing trajectory difficulty for all methods. Nevertheless, EPiC consistently outperforms all baselines across all difficulty levels.
>
> ||Easy|Medium|Hard|
> |---|---|---|---|
> |Method|CamMC↓| CamMC↓ | CamMC↓ |
> |CameraCtrl| 1.58| 2.13 | 3.55 |
> |AC3D| 1.30 | 1.77| 3.00 |
> |ViewCrafter| 0.87| 1.21 | 2.08 |
> |FloVD| 0.96 | 1.32 | 2.24 |
> |Gen3C| 0.85 | 1.13 | 2.25 |
> |EPiC (Ours)| **0.71** | **1.02** | **1.90** |
>
> Overall, these results confirm that EPiC is robust across a wide range of trajectory difficulties and flow estimators, consistently outperforming baselines in all settings.
>
> >**W2 & Q2 a: Systematic V2V evaluation.**
>
> We thank the reviewer for the suggestions. We provide additional evaluation and analysis on Kubric-4D.
>
> **Quantitative evaluation.** We have already provided partial V2V results on Kubric-4D in Table 2. We show full baseline comparison here:
>
> | Method | PSNR ↑ | SSIM ↑ | Trained with V2V-specific Data? |
> |---|---|---|---|
> | ReCamMaster | 18.52 | 0.55 | Yes (V2V data) |
> | GCD | 19.72 | 0.59 | Yes (Kubric train set) |
> | Gen3C | 19.69 | 0.61 | Yes (Kubric train set) |
> | TrajCrafter | 19.61 | 0.62 | Yes (V2V data) |
> | EPiC (Ours) | 19.65 | 0.60 | **No (zero-shot)** |
>
> EPiC achieves comparable or better results in a **zero-shot** setting (trained only on I2V data), while GCD/Gen3C are trained on Kubric train set, and TrajCrafter/ReCamMaster use V2V-specific training data.
>
> **Qualitative evaluation.** We provide qualitative examples at our [website](https://epic-submission.netlify.app/rebuttal_r4.html). The results show that EPiC produces comparable or better videos than baselines despite being zero-shot. We also note that our qualitative comparisons in general scenarios (Figs. 4(b), 15, 16) show EPiC better preserves semantic priors under occlusion errors, while others tend to rigidly follow erroneous anchor regions or generalize worse to moving camera cases.
>
>
> >**W2 & Q2 b: Failure analysis**
>
> We have provided analysis in Appendix I, identifying two main failure modes: (1) incorrect point-cloud structure leading to geometric distortions, and (2) incorrect occlusion causing some visual artifacts. We will surface these more prominently in the revision.
>
>
> >**W3: Baseline Fairness**
>
> We agree that clarifying baseline configs is important for fair comparison. All baselines use their officially released models with recommended settings. The backbones used are: ReCamMaster—Wan2.1 (1.3B), CameraCtrl/GCD — SVD (2.5B), AC3D/FloVD/TrajCrafter/EPiC — CogVideoX (5B), ViewCrafter—DynamiCrafter (5.6B), Gen3C — Cosmos (7B).  For anchor-video-based methods (ViewCrafter, TrajCrafter, Gen3C, and EPiC), we use the same anchor videos per test sample for fair comparison (Sec. 5.1). Training compute budgets are detailed in Table 4. We will add a summary table in the revision.
>
> >**Q3: Anchor construction overhead.**
>
> Masking-based anchor construction is training-only (inference uses point-cloud anchors with no additional overhead) and is preprocessed before training. Optical flow visibility masking takes ~2.9s/video as RAFT is very efficient, while artifact injection adds ~0.1s. Total: ~3s/video, ~4.5 hours for the full 5K training set on a single GPU.
>
>
> >**Q4: ControlNet placement sensitivity**
>
> Table 7 ablates the number of controlnet injection layers (2, 8, 21 layers): 8 layers provide the best balance. Regarding visibility-mask fusion: Figure 9 ablates VAOM at training vs. inference time, showing that applying VAOM during both training and inference is critical. Regarding sampler/step sensitivity: this is a backbone property — we follow the standards to randomly sample timesteps during training, so the model is not tied to a specific sampler or step count. At inference we follow the backbone's default settings (DDIM, 50 steps).

---

> > ### Author Rebuttal · Reviewer_hbsf · 2026-03-31
> >
> > Thank you for the detailed rebuttal. The additional analysis addresses my main concerns. The added Kubric-4D comparison and failure analysis make the zero-shot V2V claim more convincing, even if that part of the paper remains less extensive than the main I2V evaluation. I also appreciate the clarifications on robustness, fairness, and overhead.

---

### Official Review · Reviewer_Xaxx · 2026-03-10

**Soundness:** 3
**Presentation:** 3
**Significance:** 4
**Originality:** 4
**Overall Recommendation:** 5
**Confidence:** 2

**Summary:**

This paper studies controllable image-to-video and video-to-video generation with explicit camera motion control. The authors investigate a central concept: whether camera control can be learned more efficiently by giving the model a cleaner and better aligned guidance signal, instead of relying on noisy 3D reconstructions and estimated camera poses. Their main idea is to replace conventional point-cloud-rendered training anchors with visibility-masked anchor videos built directly from source videos using first-frame visibility tracked by optical flow. This produces anchors that stay well aligned in visible regions while naturally masking newly revealed content, which makes the learning problem much simpler. On top of that, they introduce a lightweight Anchor-ControlNet that injects these anchor-video signals into a frozen video diffusion backbone only in visible regions, so the control module copies trustworthy content while the base model handles unseen regions.

Overall, the submission shows that precise camera control does not necessarily require heavy 3D preprocessing, large-scale backbone finetuning, or massive training budgets. EPiC is presented as a much more efficient alternative: it trains with a small adapter, far fewer videos, and very few optimization steps, yet still reports strong camera accuracy and competitive visual quality on RealEstate10K and MiraData for image-to-video camera control, while also generalizing zero-shot to video-to-video settings. The paper also extends the framework to practical variants such as masked point clouds for dynamic foreground/background motion, regional animation, and object-level trajectory manipulation, making the method feel like a general anchor-guided control framework rather than only a narrow camera-control trick.

**Compliance With Llm Reviewing Policy:**

Affirmed.

**Final Justification:**

The authors fully addressed my concerns. I decided to maintain my score.

**Key Questions For Authors:**

1. How much of the gain comes specifically from better anchor alignment, versus from the lightweight frozen-backbone ControlNet design?
The paper’s central narrative is that anchor-source misalignment is the main bottleneck in prior work, but the current evaluation only partially separates this from architectural and training-budget differences. A more controlled comparison under matched backbone, data, and optimization budget would help clarify this point. If the authors can show that anchor alignment itself is the dominant factor, that would further strengthen my confidence in the main causal claim of the paper.

2. Can the authors provide a more direct quantitative analysis of anchor quality, such as an alignment or correspondence metric between anchor and source videos, and relate it to downstream performance?
Right now, the argument that masking-based anchors are superior is supported mainly through final generation metrics and qualitative examples. A direct anchor-quality analysis would make the paper’s core mechanism much more convincing.

**Limitations:**

No. The paper does include both an impact statement and a short limitations section, and it appropriately notes two relevant points: EPiC depends on the capabilities of the underlying video diffusion backbone, and like other video generation methods it could be misused to create misleading content.
However, the discussion is too brief for this paper. In particular, the authors should more explicitly discuss method-specific limitations that are already visible elsewhere in the paper, such as sensitivity to errors in point-cloud structure, occlusion masks, and optical-flow-based visibility estimation, especially in dynamic scenes or heavy occlusion cases. The appendix failure analysis already suggests these are real possible failures, so they should be surfaced more clearly in the main limitations discussion.

**Strengths And Weaknesses:**

The authors investigate a central concept in controllable video generation: whether camera control can be learned more efficiently by replacing noisy 3D-rendered anchors with better aligned, visibility-based anchor videos, and by confining control to regions where alignment is trustworthy. Overall, the submission contributes notably by reframing anchor-video conditioning as an alignment problem rather than a pure model-capacity problem, and then building a lightweight solution around that view. The paper is strong in motivation and has a clear method story, with some minor concerns that the empirical case does not fully prove the stronger causal claims the paper sometimes suggests.

**Soundness:**
The paper is technically plausible and mostly well executed. The proposed pipeline is coherent: construct anchor videos by tracing first-frame visibility with optical flow, then use a small visibility-aware Anchor-ControlNet to inject only trustworthy visible-region signals while leaving invisible regions to the frozen base model. This is a reasonable design, and the ablations are aligned with the paper’s core claims. In particular, the comparison between masking-based anchors and point-cloud-based anchors is important and supports the main intuition that better alignment makes learning easier. The main quantitative results on RE10K and MIRA are also strong: EPiC appears to outperform or match prior methods on camera metrics while using dramatically less training data, fewer optimization steps, and far fewer trainable parameters.

That said, the soundness story is not completely airtight. The paper argues that anchor misalignment is the main reason prior anchor-based methods are hard to train and computationally heavy, but the evaluation only partially isolates this claim. The observed gains could come from several factors at once: better alignment, easier supervision, the frozen-backbone ControlNet design, or the specific training setup. The evidence is supportive, but not fully causal. I also found the V2V evidence weaker than the I2V evidence: the paper emphasizes zero-shot generalization, but the quantitative V2V evaluation appears narrower and less decisive than the main I2V results. More controlled comparisons under matched backbones, matched training budgets, and direct measurements of anchor-source alignment error would make the technical case stronger. Overall, I think the paper is sound enough for serious consideration, but some of its broader claims are better viewed as well-supported hypotheses.

**Presentation:**
The paper is clearly written and easy to follow. The narrative is one of its strengths: prior methods depend on 3D-based anchors; those anchors are often misaligned; misalignment makes learning harder; therefore one should construct better aligned anchors directly from the source video and use a lightweight conditioning path that only acts where the anchor is trustworthy. That progression is intuitive, and the method components map cleanly onto the stated motivation. The figures and qualitative examples also help explain the intended behavior, especially the distinction between visible regions that should be copied and invisible regions that should be generated by the backbone.

The main presentation weakness is that the paper occasionally overstates the simplicity of the problem after reformulation. Phrases suggesting that the task becomes mostly “copying visible regions” underplay the fact that camera control still requires plausible synthesis of unseen areas, temporal consistency, and handling dynamic scenes. I would encourage the authors to be slightly more careful and explicit about what their method solves well versus what is still delegated to the pretrained backbone. I also think the paper could do a better job separating “evidence that EPiC works” from “evidence that misalignment is the dominant bottleneck in prior work.” Those are related but not identical claims. A more explicit discussion of those distinctions would improve the paper’s precision.

**Significance:**
The paper addresses a meaningful problem. Camera control is an important capability for video generation, and current approaches often appear expensive, fragile, or tied to 3D pipelines that are difficult to scale to real-world video. A method that reduces training cost by a large margin while maintaining or improving camera accuracy is practically relevant. Even if the contribution is somewhat specialized, it is specialized in a useful place: controllable generation is becoming central to how video models are used, and reducing reliance on costly 3D preprocessing could make such methods easier to train and adapt. The fact that EPiC reportedly trains with only 5K videos, 500 steps, and roughly 15 GPU hours gives the work real practical appeal beyond benchmark gains.

The paper's significance comes from showing a more efficient formulation of an already important problem. If the results hold up, this could influence how future camera-control methods think about supervision quality and how they divide responsibility between explicit control signals and the generative backbone.

**Originality:**
The paper is reasonably original. None of the individual ingredients is completely unprecedented on its own: masking based on visibility, lightweight control modules with frozen backbones, and restricting conditioning to more reliable regions are all ideas that fit within existing trends. What feels novel here is the way these ideas are combined around a specific insight: anchor videos should only supervise what can be aligned well, and visible/invisible regions should be treated differently during conditioning. The method also removes a fairly restrictive assumption in prior work, namely the need to depend heavily on 3D point clouds or camera trajectory estimation for building anchors.

**Overall Assessment:**
The paper has a strong and accessible motivation, a clean method design, and a good match between the stated problem and the proposed solution. The efficiency gains are compelling on paper, and the I2V quantitative results appear strong. The ablations are better than average because they target the paper’s central design choices rather than peripheral details.
The main weakness is that the evaluation does not fully prove the strongest version of the paper’s storyline. It shows that EPiC works and that its design choices help, but it does not completely isolate misalignment as the key reason prior methods are inefficient. I also think the paper occasionally frames its contribution a bit too absolutely, especially when contrasting its approach with all prior 3D-based methods.

---

> ### Author Rebuttal · Authors · 2026-03-30
>
> We sincerely thank Reveiwer Xaxx for the thorough and insightful review and for recognizing the significance and originality of our contribution.
>
> >**Q1: Disentangling alignment quality from architectural design**
>
> We agree that this is an important question. Our ablation in Table 3 directly isolates the anchor type while keeping the architecture identical (same Anchor-ControlNet, same backbone, same training setup):
>
> - **Same architecture + point-cloud anchors** (1500 iter): RotErr 0.60, TransErr 1.07, CamMC 1.45
> - **Same architecture + masking-based anchors** (500 iter): RotErr 0.40, TransErr 0.86, CamMC 1.17
>
> This comparison differs only in anchor construction. The masking-based anchors achieve substantially better results with 3× fewer iterations, directly isolating the anchor alignment effect.  Beyond this, Table 7 shows that with our well-aligned masking-based data, even a very minimal ControlNet (no pretraining, 256 hidden dim, 8 layers) achieves strong performance, directly enhancing the data alignment benefit.
>
> >**Q2: Direct anchor quality metric**
>
> Thanks for this question. We measured anchor–source alignment quality (PSNR in visible regions) and its correlation with downstream camera control performance on RE10K.
>
> | Training Anchor (RE10K) | Anchor PSNR ↑ | RotErr (↓) | TransErr (↓) | CamMC (↓) |
> |---|---|---|---|---|
> | Point-cloud-based (1500 iter) | 16.01 | 0.60±0.20 | 1.07±0.39 | 1.45±0.62 |
> | Mixed 50% mask + 50% PC (1500 iter) | 28.07 | 0.52±0.15 | 0.99±0.24 | 1.29±0.32 |
> | Masking-based (ours, 500 iter) | 40.12 | 0.43±0.10 | 0.84±0.22 | 1.06±0.25 |
>
> The results clearly show that better anchor alignment directly leads to more efficient and effective training: higher anchor PSNR correlates with lower camera errors, and our masking-based approach achieves the best results with 3x fewer iterations. Will add to the revised paper.
>
>
> >**Presentation: Overstating simplicity of "copying visible regions"**
>
> We appreciate this feedback and will revise the language. We agree that camera control still requires plausible synthesis of unseen areas, temporal consistency, and handling of dynamic scenes — all of which are handled by the **frozen base diffusion model**. Our contribution is specifically in making the ControlNet's task simpler (copying well-aligned visible content), which in turn allows the base model's generative capabilities to be fully preserved for the harder problem of invisible-region synthesis. We will make this division of responsibility more explicit.
>
> >**Limitations discussion**
>
> We will expand the limitations section to explicitly discuss: (1) sensitivity to optical flow errors in fast-motion/heavy-occlusion scenes, (2) the "freeze mask" heuristic and its failure modes under extreme viewpoint changes, and (3) dependence on the quality of test-time depth estimation for point-cloud rendering. We note that our failure analysis in Appendix I already identifies two concrete failure modes (incorrect point-cloud structure and incorrect occlusion), and we will surface these more prominently in the main text.

---

> > ### Author Rebuttal · Reviewer_Xaxx · 2026-03-31
> >
> > I appreciate the authors' response. I do not have any further concerns.

---

### Official Review · Reviewer_2E8C · 2026-03-13

**Soundness:** 3
**Presentation:** 2
**Significance:** 3
**Originality:** 2
**Overall Recommendation:** 4
**Confidence:** 2

**Summary:**

This manuscript presents an efficient and precise camera-controlled video generation framework based on video diffusion models. The authors propose a novel strategy to construct well-aligned training anchor videos without requiring camera pose estimation or point-cloud reconstruction. Specifically, they estimate optical flow from the first frame to each subsequent frame and mask out invisible pixels based on the flow, producing anchor videos that remain well aligned in visible regions. To improve the generalization of a model trained with these anchors and to reduce the train–test discrepancy, the authors further inject “flying-pixel” artifacts into the training anchors to better match the anchor distribution at inference time. The constructed anchor video is then encoded and injected into the main video diffusion model via a lightweight ControlNet. During inference, anchor videos are rendered from estimated point clouds. In addition, the method supports video-to-video inference without additional training. Experimental results show that the proposed framework EPiC achieves competitive quality compared with existing methods while being more efficient.

**Compliance With Llm Reviewing Policy:**

Affirmed.

**Key Questions For Authors:**

Refer to the weakness

**Limitations:**

Refer to the weakness

**Strengths And Weaknesses:**

Strength
1. The proposed framework constructs conditioning signals without requiring camera pose parameters or point clouds during training, providing a simple yet effective way to build large-scale datasets for camera-controlled video generation.

2. Training is efficient and lightweight, making the approach practical under limited computational resources.

Weakness
1. Although the proposed anchor construction pipeline injects artifacts to reduce the gap between training and inference, there may still remain a mismatch between the two settings. To better validate the effectiveness of the proposed design, I recommend an ablation where the authors use the inference-time anchor construction strategy (i.e., point-cloud-rendered anchors) during training, or a mixed training setting that includes such anchors.

2. The model is trained with a relatively small-scale dataset and limited iterations. Do the authors have results trained on a larger dataset and/or with more training iterations? Such experiments would better demonstrate the scalability and robustness of the proposed data construction method, especially since it avoids pose estimation and point-cloud extraction during training.

3. I also recommend validating the framework on at least one additional base video diffusion model to better demonstrate that the proposed strategy generalizes beyond a specific backbone.

---

> ### Author Rebuttal · Authors · 2026-03-30
>
> We thank the reviewer for recognizing the simplicity and effectiveness of our dataset construction pipeline and the efficiency of our training approach, as well as the constructive suggestions.
>
> >**W1: Train–test mismatch; ablation with point-cloud anchors during training.**
>
> We thank the reviewer for this great suggestion. We note that this ablation is only feasible on datasets with camera annotations (e.g., RE10K), as Panda70M has no such annotations for point-cloud rendering. We already have related experiments: Table 3 compares point-cloud-based vs. masking-based anchor construction under the same architecture, and Table 5 evaluates masking-based training with RE10K data source. We summarize them here with an additional mixed-anchor setting (all evaluated on RE10K dataset):
>
> | Training Anchor (RE10K) | RotErr (↓) | TransErr (↓) | CamMC (↓) |
> |---|---|---|---|
> | Point-cloud-based (iter 1500) | 0.60±0.20 | 1.07±0.39 | 1.45±0.62 |
> | Mixed 50% mask + 50% Point-cloud (iter 1500) | 0.52±0.15 | 0.99±0.24 | 1.29±0.32 |
> | Masking-based (iter 500) | 0.43±0.10 | 0.84±0.22 | 1.06±0.25 |
>
>
> Ours (Masking-based anchors) achieve the best results with the fewest iterations thanks to strong anchor-target alignment. Adding point-cloud anchors into the mix hurts performance; we attribute this to mixing introducing misalignment additionally. Full point-cloud-based training performs the worst, yet still yields higher errors even though baselines were trained for 3x more steps due to systematic misalignment throughout visible regions.
>
> >**W2: Scalability to larger data and training regime**
>
> We agree that exploring how performance evolves with increased training scale is interesting. We scaled training data to 30K samples from Panda70M and trained for 5K steps. As shown below (same test sets as Table 1), EPiC (scale up) improves over EPiC:
>
> | | #Videos | Visual Quality | RotErr (↓) | TransErr (↓) | CamMC (↓) |
> |---|---|---|---|---|---|
> | EPiC on RE10K | 5K | 82.63 | 0.40±0.11 | 0.86±0.18 | 1.17±0.23 |
> | EPiC (scale up) on RE10K | 30K | 82.70 | 0.34±0.10 | 0.83±0.20 | 1.01±0.21 |
> | EPiC on MiraData | 5K | 82.89 | 0.66±0.22 | 1.78±0.67 | 2.10±0.60 |
> | EPiC (scale up) on MiraData | 30K | 82.91 | 0.60±0.20 | 1.69±0.61 | 2.01±0.57 |
>
> Note that EPiC already achieves SoTA with only 5K videos and 500 steps (Table 1, Table 4), which is over an order of magnitude less than prior methods. Scaling up further improves results, demonstrating that our framework benefits from more data. We believe the strong baseline performance with minimal resources is a key advantage of our well-aligned training pipeline: it enables fast convergence (as shown by the training curves in Fig. 5(a)), while still leaving room for improvement with additional scale.
>
>
> >**W3: Generalization across base models**
>
> We thank the reviewer for this suggestion. We’d like to bring your attention to our experiments on RE10K with another backbone — Wan2.1-I2V-14B in Appendix D.5, Table 6, and Fig. 10.  Results:
>
> | Method (Backbone) | Total (VBench) | RotErr (↓) | TransErr (↓) | CamMC (↓) |
> |---|---|---|---|---|
> | CameraCtrl (SVD 2.5B) | 78.35 | 1.12±0.44 | 1.78±0.93 | 2.36±1.01 |
> | AC3D (CogVideoX 5B) | 82.63 | 0.86±0.37 | 1.50±0.82 | 1.97±0.86 |
> | ViewCrafter (DynamiCrafter 5.6B) | 81.18 | 0.50±0.16 | 1.05±0.32 | 1.35±0.40 |
> | FloVD (CogVideoX 5B) | 82.61 | 0.76±0.31 | 1.14±0.52 | 1.47±0.56 |
> | Gen3C (Cosmos 7B) | 82.27 | 0.45±0.13 | 0.99±0.18 | 1.35±0.30 |
> | EPiC (CogVideoX 5B) | **82.63** | **0.40±0.11** | **0.86±0.18** | **1.17±0.23** |
> | EPiC (Wan2.1 14B) | **84.24** | **0.41±0.10** | **0.84±0.20** | **1.15±0.21** |
>
>
>
> The Wan2.1-14B backbone yields higher visual quality with comparable camera accuracy, demonstrating that EPiC generalizes well across different backbones. Notably, even the smaller CogVideoX-5B backbone equipped with EPiC already outperforms all baselines (including those built on larger backbones) in camera control accuracy, suggesting that our well-aligned training pipeline is the primary driver of performance rather than backbone capacity.

---

> > ### Author Rebuttal · Reviewer_2E8C · 2026-04-03
> >
> > I appreciate the authors' rebuttal, and keep my rating.

---

### Official Review · Reviewer_aBVH · 2026-03-13

**Soundness:** 3
**Presentation:** 3
**Significance:** 2
**Originality:** 3
**Overall Recommendation:** 4
**Confidence:** 4

**Summary:**

This paper proposes EPiC, a camera-control framework for video diffusion models that replaces point-cloud/camera-pose-based training anchors with visibility-masked anchor videos constructed directly from source videos. The method also introduces a lightweight Anchor-ControlNet with visibility-aware output masking, so control is applied only to visible regions while disoccluded content is left to the frozen base model. Experiments on RealEstate10K and MiraData show improved camera-control accuracy over several recent baselines, with much lower training cost, fewer trainable parameters, and fewer training videos. The paper also presents zero-shot transfer to V2V camera control.

**Compliance With Llm Reviewing Policy:**

Affirmed.

**Final Justification:**

Thank you for the rebuttal. Some of my concerns were caused by unclear presentation. I suggest the authors clarify these potentially misleading parts in the final version. I will raise my score to Weak Accept.

**Key Questions For Authors:**

1. How sensitive is the training-anchor quality to the choice of optical flow model? Would performance degrade significantly with a weaker or faster flow estimator?

2. It is unclear why the training data is chosen from Panda-70M rather than datasets such as RealEstate10K or DL3DV, which are commonly used in camera-control or view-synthesis research. Is the intention to emphasize that the proposed method does not require datasets with camera pose annotations? If so, this motivation should be stated more explicitly, and the implications for training data requirements should be discussed more clearly.

**Limitations:**

Please refer to the weaknesses.

**Strengths And Weaknesses:**

* Strengths

1. Clear problem diagnosis. The paper argues that prior anchor-video methods suffer from visible-region misalignment due to inaccurate point clouds and camera trajectories, which increases optimization difficulty. This framing is convincing and consistent with the qualitative evidence.

2. Compelling ablations. The comparison between point-cloud-based anchors and masking-based anchors is one of the strongest parts of the paper; it directly validates the paper’s core claim that alignment quality matters more than explicit 3D rendering during training.

3. The results seems better than previous methods.

* Weaknesses:

1. Fundamentally, the method does not address the core issue of multi-frame estimation errors. Instead, it effectively shifts the source of error from camera and depth estimation to optical flow estimation. It remains unclear whether the flow model is sufficiently accurate or robust across diverse scenes. Overall, the approach does not resolve the fundamental challenges in this area, but rather replaces existing components with another module that appears to perform better empirically.

2. The paper may underplay dependence on 3D at test time. A central selling point is that training avoids camera pose and point cloud estimation. However, for I2V inference the method still relies on depth estimation, point-cloud rendering, and optionally segmentation for dynamic-object masking; V2V inference also depends on estimated dynamic depths/point clouds. So the method removes 3D dependence primarily from training, not from the full pipeline.

3. The evaluation mainly relies on VLM-based scoring metrics, while traditional image/video fidelity metrics such as PSNR, SSIM, and LPIPS are not reported.

---

> ### Author Rebuttal · Authors · 2026-03-30
>
> We thank the reviewer for the thoughtful feedback and for recognizing our formulation, ablations, and empirical gains. We address concerns on error sources, 3D dependence, evaluation metrics, and data design below.
>
> >**W1: 3D estimation errors**
>
> We acknowledge that our method does not directly fix multi-frame estimation errors, instead, we show that **fixing them can be unnecessary for constructing well-aligned training anchor videos**. The key insight is to sidestep the 3D estimation-and-rendering pipeline, 3D estimation-and-rendering pipeline, which often introduces anchor-source misalignment, via visibility-based masking when constructing anchor videos. This enforces strong condition–target alignment in visible regions without requiring accurate 3D estimation at all.
>
> Concretely, in prior methods (Gen3C, TrajCrafter, ViewCrafter etc), the 3D-estimation-to- rendering pipeline often introduces anchor–target misalignment due to estimation errors. Our approach avoids this, as the anchor is constructed by directly copying source pixels in visible regions and masking out the rest. Optical flow here is used only to estimate the visibility mask, not as a conditioning signal. Even with imperfect flow, the errors only affect mask boundaries and do not alter visible-region content, so pixel alignment within visible regions is always preserved by construction. Thus, estimation errors will have limited impact on learning camera control: since the training task is mainly visible-region copy-pasting, supervision is restricted to such aligned visible regions, making the method robust to flow inaccuracies.
>
> Empirically, our masking-based approach achieves RotErr 0.40 vs. 0.60 and TransErr 0.86 vs. 1.07 with 3x fewer iterations compared to point-cloud-based anchors (Table 3). See also Q1 below for flow model sensitivity analysis.
>
> >**W2: 3D dependence at test time.**
>
> We thank the reviewer for raising this important point. Our intention is to follow the well-established 3D-informed anchor video paradigm (as discussed in L012–017, L023–026), where all methods in this line of work, including ours, rely on 3D at test time. Our primary focus is on addressing training-time anchor misalignment, and we agree that our writing may have over-emphasized this point, giving the impression of downplaying 3D dependence. We will revise to better balance the presentation.
>
> >**W3: Missing PSNR/SSIM/LPIPS metrics**
>
> We appreciate the reviewer for the thoughtful comments. We performed additional evaluation with the mentioned metrics on RealEstate10K (MiraData does not have ground-truth novel views for reference-based evaluation). We report results on the full sampled test set from Table 1, and also sample an easy subset where camera rotation < 10° and translation < 0.5 units:
>
> | | Easy set | | | Full set | | |
> |---|---|---|---|---|---|---|
> | Method | PSNR ↑ | SSIM ↑ | LPIPS ↓ | PSNR ↑ | SSIM ↑ | LPIPS ↓ |
> | CameraCtrl | 15.34 | 0.591 | 0.391 | 12.06 | 0.501 | 0.509 |
> | AC3D | 18.34 | 0.615 | 0.233 | 14.30 | 0.581 | 0.402 |
> | FloVD | 18.52 | 0.621 | 0.228 | 14.45 | 0.583 | 0.398 |
> | ViewCrafter | 19.36 | 0.660 | 0.200 | 14.91 | 0.579 | 0.386 |
> | Gen3C | **19.93** | 0.667 | **0.180** | 15.42 | 0.587 | 0.372 |
> | EPiC (Ours) | 19.91 | **0.671** | 0.184 | **15.51** | **0.594** | **0.374** |
>
> Our model achieves comparable or better results against baselines. Note that Gen3C performs comparably to ours, which is expected as it also uses anchor-video-based guidance, while ours are much more efficient. We will add this table in the revision.
>
>
> >**Q1: Optical flow model sensitivity.**
>
> Thanks for the insightful question. We tested three different flow models:
>
> | Flow Model | RotErr (↓) | TransErr (↓) | CamMC (↓) |
> |---|---|---|---|
> | RAFT | 0.40±0.11 | 0.86±0.18 | 1.17±0.23 |
> | UniMatch [1] | 0.42±0.12 | 0.85±0.19 | 1.19±0.25 |
> | GMFlow [2] | 0.40±0.13 | 0.88±0.21 | 1.19±0.27 |
>
> Results are largely similar across optical flow models, reinforcing that **masking itself is the key factor**. Since flow errors mainly affect mask boundaries, different estimators still produce well-aligned visible regions for efficient learning. We will add this analysis to the revision.
>
> [1] Unifying Flow, Stereo and Depth Estimation, Xu et al., 2022.
>
> [2] GMFlow: Learning Optical Flow via Global Matching, Xu et al., 2021.
>
>
> >**Q2: Choice of Panda70M for training data**
>
> We clarify that this is precisely our motivation: EPiC does not require (high-quality) camera pose annotations, enabling training on arbitrary video data. We choose Panda70M for its diverse, in-the-wild dynamic scenes, whereas RE10K/DL3DV are limited to largely static indoor content. This is supported by our ablation (Table 5, Fig. 7, Appx. D.1): training on Panda70M achieves comparable camera accuracy on static scenes (RealEstate10K), while significantly improving camera following in dynamic scenes (higher MiraData accuracy and better V2V generalization). We will clarify this motivation in the revision.

---

> > ### Author Rebuttal · Reviewer_aBVH · 2026-04-03
> >
> > Thank you for your response. After careful consideration, I feel that the methodology effectively translates the reconstruction error into the flow estimation domain. While technically sound, it feels more like a redistribution of error rather than a substantial reduction of the problem's inherent difficulty. I have decided to keep my score unchanged.

---

> > > ### Author Response · Authors · 2026-04-04
> > >
> > > Thanks for the follow-up and for acknowledging that our method is technically sound. We address the remaining concerns below.
> > >
> > > > **Regarding "The problem's inherent difficulty"**
> > >
> > > If we understand correctly, for "the problem's inherent difficulty", the reviewer refers to "multi-frame 3D estimation/reconstruction errors", as highlighted by the reviewer in the first round. If so, we believe there may be a misunderstanding regarding the problem our work aims to address.
> > > We agree that these errors are a fundamental challenge if the goal is to train a 3D estimator/reconstructor (e.g., VGGT). However, this is not our goal. We focus on learning anchor-video-following for **video generative model camera control**, where 3D estimators serve **merely as a tool**  for producing anchor videos (similar to Gen3C, TrajCrafter, ViewCrafter, etc.), not as optimization targets. In this context of learning anchor-video following, we argue that the inherent difficulty is instead anchor–target misalignment, which we found directly impacts how efficiently and accurately the model learns to follow the anchor. Our method addresses this by enforcing strong alignment in visible regions via masking, bypassing 3D estimation errors entirely.
> > >
> > > To quantify this, we measured anchor–source alignment (PSNR in visible regions) and its correlation with downstream performance on RE10K:
> > >
> > > | Training Anchor | Anchor PSNR ↑ | RotErr (↓) | TransErr (↓) | CamMC (↓) |
> > > |---|---|---|---|---|
> > > | Point-cloud-based (1500 iter) | 16.01 | 0.60±0.20 | 1.07±0.39 | 1.45±0.62 |
> > > | Masking-based (ours, 500 iter) | 40.12 | 0.43±0.10 | 0.84±0.22 | 1.06±0.25 |
> > >
> > > The anchor PSNR gap (16.01 → 40.12) also demonstrates that this is not a redistribution of error, but a fundamental improvement in anchor quality, which directly translates to better and more efficient camera control learning.
> > >
> > > > **Regarding "Redistribution of error to optical flow"**
> > >
> > > We'd like to point out that the reviewer's observation applied to a prior work, but not ours — specifically FloVD (Jin et al. 2025), which we discussed in our paper (L152–156 and L969–973). As we discussed, FloVD directly uses raw estimated optical flow as a conditioning signal to learn camera control, and still suffers from condition–target misalignment due to flow estimation errors, despite replacing 3D-based anchor videos with flow, a case that truly fits the "error redistribution from 3D estimation to optical flow" characterized by the reviewer.
> > >
> > > However, our method is fundamentally different: we do not condition on optical flow — we still condition on anchor videos, but with substantially better alignment than prior work. Optical flow estimator is solely a tool for estimating the visibility mask, and the anchors are constructed by directly copying source pixels in visible regions. Flow errors may affect mask boundaries, but do not affect alignment within visible regions, which guarantees strong alignment regardless of flow accuracy (anchor PSNR 40.12 vs 16.01 for point-cloud-based anchors). Thus, instead of a redistribution of error, what we achieved is an elimination of visible-region misalignment, which is the core bottleneck in prior anchor-based methods. This is reflected in our results, where EPiC substantially outperforms FloVD (Table 1) and other anchor-video-based methods.
> > >
> > > We hope this clarifies our perspective and are happy to discuss further.

---

### Decision · Program_Chairs · 2026-04-30

**Decision:**

Accept (regular)

**Comment:**

This paper received unanimously positive reviews: 4, 4, 5, 5. The reviewers find that the paper makes a convincing case that pointcloud-based pose control for video generators is overly sensitive to pointcloud errors, and that the proposed approach based on optical flow and binary masking is better, both qualitatively and quantitatively. The reviewers note that the efficiency claims are "unusually strong", backed by clear and convincing evidence. Based on this positive consensus, the AC recommends to accept. In preparing the final copy, the authors are encouraged to take seriously the closing remarks from the reviewers, including the comment from Xaxx that "the paper occasionally frames its contribution a bit too absolutely, especially when contrasting its approach with all prior 3D-based methods" -- correcting such over-claims will strengthen the work.